# Editing a Classifier by Rewriting Its Prediction Rules

**Shibani Santurkar**[*]
MIT
shibani@mit.edu

**Dimitris Tsipras**[*]
MIT
tsipras@mit.edu

**Mahalaxmi Elango**
MIT
melango@mit.edu

**David Bau**
MIT
davidbau@mit.edu

**Antonio Torralba**
MIT
torralba@mit.edu

**Aleksander Mądry**
MIT
madry@mit.edu

## Abstract

We present a methodology for modifying the behavior of a classifier by *directly rewriting* its prediction rules.[1] Our approach requires virtually no additional data collection and can be applied to a variety of settings, including adapting a model to new environments, and modifying it to ignore spurious features.

## 1 Introduction

At the core of machine learning is the ability to automatically discover prediction rules from raw data. However, there is mounting evidence that not all of these rules are reliable [Torralba and Efros, 2011, Beery et al., 2018, Shetty et al., 2019, Agarwal et al., 2020, Xiao et al., 2020, Bissoto et al., 2020, Geirhos et al., 2020]. In particular, some rules could be based on biases in the training data: e.g., learning to associate cows with grass since they are typically depicted on pastures [Beery et al., 2018]. While such prediction rules may be useful in some scenarios, they will be irrelevant or misleading in others. This raises the question:

*How can we most effectively modify the way in which a given model makes its predictions?*

The canonical approach for performing such post hoc modifications is to intervene at the data level. For example, by gathering additional data that better reflects the real world (e.g., images of cows on the beach) and then using it to further train the model. Unfortunately, collecting such data can be challenging: how do we get cows to pose for us in a variety of environments? Furthermore, data collection is ultimately a very indirect way of specifying the intended model behavior. After all, even when data has been carefully curated to reflect a given real-world task, models still end up learning unintended prediction rules from it [Ponce et al., 2006, Torralba and Efros, 2011, Tsipras et al., 2020, Beyer et al., 2020].

**Our contributions**

The goal of our work is to develop a toolkit that enables users to *directly modify* the prediction rules learned by an (image) classifier, as opposed to doing so implicitly via the data. Concretely:

**Editing prediction rules.** We build on the recent work of Bau et al. [2020a] to develop a method for modifying a classifier's prediction rules with essentially *no* additional data collection (Section 2). At a high level, our method enables the user to modify the weight of a layer so that the latent

---

[*]Equal contribution.

[1]Our code is available at https://github.com/MadryLab/EditingClassifiers.

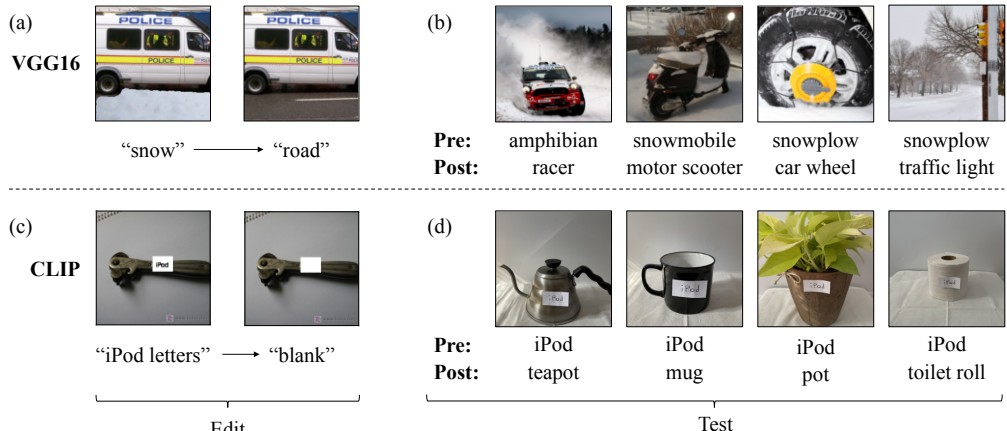

Figure 1: Editing prediction rules in pre-trained classifiers using a *single* exemplar. (a) We edit a VGG16 ImageNet classifier to map the representation of the concept "snow" to that of "asphalt road". (b) This edit corrects classification errors on snowy scenes corresponding to various classes. (c) We edit a CLIP [Radford et al., 2021] model such that the text "iPod" maps to a blank area. (d) This change makes the model robust to the typographic attacks from Goh et al. [2021].

representations of a specific concept (e.g., snow) map to the representations of another (e.g., road). Crucially, this allows us to change the behavior of the classifier on all occurrences of that concept, beyond the specific examples (and the corresponding classes) used in the editing process.

**Real-world scenarios.** We demonstrate our approach in two scenarios motivated by real-world applications (Section 3). First, we focus on adapting an ImageNet classifier to a new environment: recognizing vehicles on snowy roads. Second, we consider the recent "typographic attack" of Goh et al. [2021] on a zero-shot CLIP [Radford et al., 2021] classifier: attaching a piece of paper with "iPod" written on it to various household items causes them to be incorrectly classified as "iPod." In both settings, we find that our approach enables us to significantly improve model performance, using only a *single, synthetic* example to perform the edit—cf. Figure 1.

**Large-scale synthetic evaluation.** To evaluate our method at scale, we develop an automated pipeline to generate a suite of varied test cases (Section 4). Our pipeline revolves around identifying specific concepts (e.g., "road" or "pasture") in an existing dataset using pre-trained instance segmentation models and then modifying them using style transfer [Gatys et al., 2016] (e.g., to create "snowy road"). We find that our editing methodology is able to consistently correct a significant fraction of model failures induced by these transformations. In contrast, standard fine-tuning approaches are unable to do so given the same data, often causing more errors than they are fixing.

**Probing model behavior with counterfactuals.** Beyond model editing, our concept-transformation pipeline can also be viewed as a scalable way of generating image counterfactuals. In Section 5, we demonstrate how such counterfactuals can be useful to gain insights into how a given model makes its predictions and pinpoint certain spurious correlations that it has picked up.

## 2   A toolkit for editing prediction rules

It has been widely observed that models pick up various context-specific correlations in the data— e.g., using the presence of "road" or a "wheel" to predict "car" (cf. Section 5). Such unreliable *prediction rules* (dependencies of predictions on specific input concepts) could hinder models when they encounter novel environments (e.g., snow-covered roads), and confusing or adversarial test conditions (e.g., cars with wooden wheels). Thus, a model designer might want to modify these rules before deploying their model.

The canonical approach to modify a classifier post hoc is to collect additional data that captures the desired deployment scenario, and use it to retrain the model. However, even setting aside the

challenges of data collection, it is not obvious a priori how much of an effect such retraining (e.g., via fine-tuning) will have. For instance, if we fine-tune our model on "cars" with wooden wheels, will it now recognize "scooters" or "trucks" with such wheels?

The goal of this work is to instead develop a more *direct* way to modify a model's behavior: rewriting its prediction rules in a targeted manner. For instance, in our previous example, we would ideally be able to modify the classifier to correctly recognize *all* vehicles with wooden wheels by simply teaching it to treat *any* wooden wheel as it would a standard one. Our approach is able to do exactly this. However, before describing this approach (Section 2.2), we first provide a brief overview of recent work by Bau et al. [2020a] which forms its basis.

## 2.1 Background: Rewriting generative models

Bau et al. [2020a] developed an approach for rewriting a deep generative model: specifically, enabling a user to replace all occurrences of one selected object (say, "dome") in the generated images with another (say, "tree"), without changing the model's behavior in other contexts. Their approach is motivated by the observation that, using a handful of example images, we can identify a vector in the model's representation space that encodes a specific high-level concept [Kim et al., 2018, Bau et al., 2020a]. Building on this, Bau et al. [2020a] treat each layer of the model as an *associative memory*, which maps such a concept vector at each spatial location in its input (which we will refer to as the *key*) to another concept vector in its output (which we will call the *value*).

In the simplest case, one can think of a linear layer with weights $W \in \mathbb{R}^{mxn}$ transforming the key $k \in \mathbb{R}^n$ to the value $v \in \mathbb{R}^m$. In this setting, Bau et al. [2020a] formulate the rewrite operation as modifying the layer weights from $W$ to $W'$ so that $v^* = W'k^*$, where $k^*$ corresponds to the old concept that we want to replace, and $v^*$ to the new concept. For instance, to replace "domes" with "trees" in the generated images, we would modify the layer so that the key $k^*$ for "dome" maps to the value $v^*$ for "tree". Consequently, when this value is fed into the downstream layers of the network it would result in a *tree* in the final image. Crucially, this update should change the model's behavior for *every* instance of the concept encoded in $k^*$—i.e., all "domes" in the images should now be "trees".

To extend this approach to typical deep generative models, two challenges remain: (1) handling non-linear layers, and (2) ensuring that the edit doesn't significantly hurt model behavior on other concepts. With these considerations in mind, Bau et al. [2020a] propose making the following rank-one updates to the parameters $W$ of an arbitrary non-linear layer $f$:

$$\min_{\Lambda} \quad \sum_{(i,j) \in S} \left\| v_{ij}^* - f(k_{ij}^*; W') \right\| \qquad \text{s.t.} \quad W' = W + \Lambda(C^{-1}d)^\top. \tag{1}$$

Here, $S$ denotes the set of spatial locations in representation space for a single image corresponding to the concept of interest, $d$ is the top eigenvector of the keys $k_{ij}^*$ at locations $(i, j) \in S$ and $C = \sum_d k_d k_d^\top$ captures the second-order statistics for other keys $k_d$. Intuitively, the goal of this update is to modify the layer parameters to rewrite the desired key-value mapping in the most minimal way. We refer the reader to Bau et al. [2020a] for further details.

## 2.2 Editing classifiers

We now shift our attention to the focus of this work: editing classifiers. To describe our approach, we use the task of enabling classifiers to detect vehicles with "wooden wheels" as a running example. At a high level, we would like to apply the approach described in Section 2.1 to modify a chosen (potentially non-linear) layer $L$ of the network to rewrite the relevant key-value association. But, we need to first determine what these relevant keys and values are.

Let us start with a single image $x$, say, from class "car", that contains the concept "wheel". Let the location of the "wheel" in the image be denoted by a binary mask $m$.[2] Then, say we have access to a transformed version of $x$—namely, $x'$—where the "car" has a "wooden wheel" (cf. Figure 2). This $x'$ could be created by manually replacing the wheel, or by applying an automated procedure such as that in Section 4. In the rest of our study, we refer to a single $(x, x')$ pair as an *exemplar*.

---

[2]Such a mask can either be obtained manually or automatically via instance segmentation (cf. Section 4).

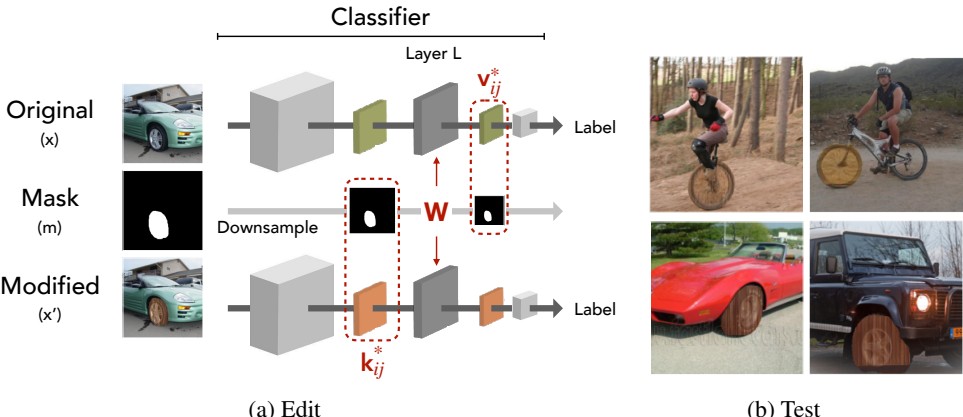

|  |  |
|---|---|
| (a) Edit | (b) Test |

Figure 2: Overview of our pipeline for directly editing the prediction-rules of a classifier. The edit in (a) seeks to modify the network to perceive wooden wheels as standard ones, using a small set of exemplar images (say from class "car"). To achieve this, we first obtain the keys $k_{ij}^*$ corresponding to the new concept (here, "wooden wheel"), and the values $v_{ij}^*$ corresponding to the original concept (here, "standard wheel") in the input and output representation space of a layer $L$ respectively. We then update the weights $W$ of the layer to enforce this new key-value association (1). (b) To test our method, we measure the improvement in model performance on test instances containing the new concept—here, images of other vehicles with "wooden wheels".

Intuitively, we want the classifier to perceive the "wooden wheel" in the transformed image $x'$ as it does the standard wheel in the original image $x$. To achieve this, we must map the keys for wooden wheels to the value corresponding to their standard counterparts. In other words, the relevant keys ($k^*$) correspond to the network's representation of the concept in the *transformed* image ($x'$) directly *before* layer $L$. Similarly, the relevant values ($v^*$) that we want to map these keys to correspond to the network's representation of the concept in the *original* images ($x$) directly *after* layer $L$. (The pertinent spatial regions in the representation space are simply determined by downsampling the mask to the appropriate dimensions.) Finally, the model edit is performed by feeding the resulting key-value pairs into the optimization problem (1) to determine the updated layer weights $W'$—cf. Figure 2 for an illustration of the overall process. Note that this approach can be easily extended to use multiple exemplars $(x_k, x_k')$ by simply expanding $S$ to include the union of relevant spatial locations (corresponding the concept of interest) across these exemplars.

## 3    Does editing work in practice?

To evaluate of our approach, we start by considering two scenarios motivated by real-world concerns: (i) adapting classifiers to handle novel weather conditions, and (ii) making models robust to typographic attacks Goh et al. [2021]. In both cases, we edit the model using a single *exemplar*, i.e., a single image that we manually annotate and modify. For comparison, we also consider two variants of fine-tuning using the same exemplar: (i) *local* fine-tuning, where we only train the weights of a single layer $L$ (similar to our editing approach); and (ii) *global* fine-tuning, where we also train all other layers between layer $L$ and the output of the model. It is worth noting that unlike fine-tuning, editing does not utilize class labels in any way. See Appendix A for experimental details.

### 3.1    Tackling new environments: Vehicles on snow

Our first use-case is adapting pre-trained classifiers to image subpopulations that are under-represented in the training data. Specifically, we focus on the task of recognizing vehicles under heavy snow conditions—a setting that could be pertinent to self-driving cars—using a VGG16 classifier trained on ImageNet-1k. To study this problem, we collect a set of real photographs from road-related ImageNet classes using Flickr (details in Appendix A.5). We then rewrite the model's prediction rules to map "snowy roads" to "road". To do so, we create an exemplar by manually annotating the concept "road" in an ImageNet image from a *different* class (here, "police van"), and the

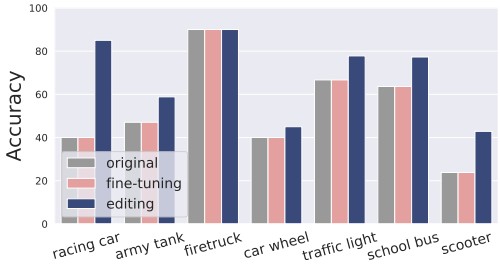
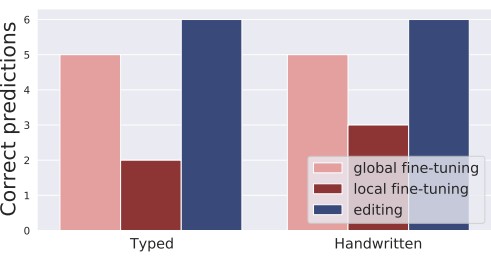

(a) Vehicles in snowy weather        (b) Typographic attacks (6 images)

Figure 3: (a) Adapting a pre-trained ImageNet VGG16 classifier to images of vehicles on snowy roads with a single exemplar. Fine-tuning (both local and global) does not improve accuracy, while editing to map "snowy road"→"road" leads to a consistent improvement across multiple classes. (b) Improving the robustness of CLIP-ResNet-50 [Radford et al., 2021] models to typographic attacks [Goh et al., 2021]. Editing the model to map the text "iPod"→"blank" using a single exemplar— either based on hand-written text on a physical teapot or from pasting typed text on an image of a "can opener"—completely corrects this vulnerability. While global fine-tuning can also improve model performance in this setting, it requires more careful hyperparameter tuning and typically hurts model performance in other contexts (Appendix Figure 9). Here, hyperparameters (cf. Appendix Table 2) are chosen based on the large-scale synthetic study in Section 4.1.

manually replace it with snow texture obtained from Flickr. We then apply our editing methodology (cf. Section 2), using this single *synthetic* snow-to-road exemplar—see Figure 1.

In Figure 3a, we measure the error rate of the model on the new test set (vehicles in snow) before and after performing the rewrite. We find that our edits significantly improve the model's error rate on these images, despite the fact that we only use a single *synthetic* exemplar (i.e., not a real "snowy road" photograph). Moreover, Figure 3a demonstrates that our method indeed changes the way that the model processes a concept (here "snow") in a way that generalizes beyond the specific class used during editing (here, the exemplar was a "police van") . In contrast, fine-tuning the model under the same setup does not improve its performance on these inputs.

One potential concern is the impact of this process on the model's accuracy on other ImageNet classes that contain snow (e.g., "ski"). On the 246 (of 50k) ImageNet test images that contain snow (identified using an MS-COCO-trained instance segmentation model [Chen et al., 2017]), the model's accuracy pre-edit is 92.27% and post-edit is 91.05%—i.e., only 3/246 images are rendered incorrect by the edit. This indicates that the classifier is not disproportionately affected by the edit.

## 3.2   Ignoring a spurious feature: Typographic attacks

Our second use-case is modifying a model to ignore a spurious feature. We focus on the recently-discovered typographic attacks from Goh et al. [2021]: simply attaching a piece of paper with the text "iPod" on it is enough to make a zero-shot CLIP [Radford et al., 2021] classifier incorrectly classify an assortment of objects to be iPods. We reproduce these attacks on the ResNet-50 variant of the model—see Appendix Figure 7 for an illustration.

To correct this behavior, we rewrite the model's prediction rules to map the text "iPod" to "blank". For the choice of our transformed input $x'$, we consider two variants: either a real photograph of a "teapot" with the typographic attack (Appendix Figure 7); or an ImageNet image of a "can opener" (randomly-chosen) with the typed text "iPod" pasted on it (Figure 1). The original image $x$ for our approach is obtained by replacing the handwritten/typed text with a white mask—cf. Figure 1. We then use this single training exemplar to perform the model edit.

In both cases, we find that editing is able to fix *all* the errors caused by the typographic attacks, see Figure 3b. Interestingly, global fine-tuning also helps to correct many of these errors (potentially by adjusting class biases), albeit less reliably (for specific hyperparameters). However, unlike editing, fine-tuning also ends up damaging the model behavior in other scenarios—e.g., the model now spuriously associates the text "iPod" with the target class used for fine-tuning and/or has lower accuracy on normal "iPod" images from the test set (Appendix Figure 9).

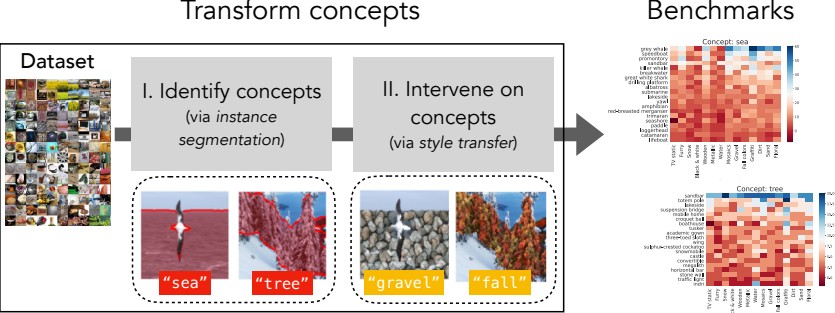

Figure 4: Creating large-scale test sets for model rewriting. Given a standard dataset, we first identify salient concepts within the corresponding images using instance segmentation, and then apply a realistic transformation to each of these concepts using style transfer [Gatys et al., 2016]. We can then evaluate the effectiveness of a model rewriting technique based on the extent to which it can alleviate the model's sensitivity (i.e., drop in accuracy) to such concept-level transformations.

# 4 Large-scale synthetic evaluation

The analysis of the previous section demonstrates that editing can improve model performance in realistic settings. However, due to the practical constraints of real-world data collection, this analysis was restricted to a relatively small test set. To corroborate the generality of our approach, we now develop a pipeline to automatically construct diverse rule-editing test cases. We then perform a large-scale evaluation and ablation of our editing method on this testbed.

## 4.1 Synthesizing concept-level transformations

At a high level, our goal is to automatically create a test set(s) in which a specific concept—possibly relevant to the detection of multiple dataset classes—undergoes a realistic transformation. To achieve this without additional data collection, we transform all instances of the concept of interest within *existing* datasets. For example, the "vehicles-on-snow" scenario of Section 3.1 can be synthetically reproduced by identifying the images within a standard dataset (say ImageNet [Deng et al., 2009, Russakovsky et al., 2015]) that contain segments of road and transforming these to make them resemble snow. Concretely, our pipeline (see Figure 4 for an illustration and Appendix A.6.1 for details), which takes as input an existing dataset, consists of the following two steps:

1. **Concept identification:** In order to identify concepts within the dataset images in a scalable manner, we leverage pre-trained instance segmentation models. In particular, using state-of-the-art segmentation models—trained on MS-COCO [Lin et al., 2014] and LVIS [Gupta et al., 2019]—we are able to automatically generate concept segmentations for a range of high-level concepts (e.g., "grass", "sea", "tree").

2. **Concept transformation:** We then transform the detected concept (within dataset images) in a consistent manner using existing methods for style transfer [Gatys et al., 2016, Ghiasi et al., 2017]. This allows us to preserve fine-grained image features and realism, while still exploring a range of potential transformations for a single concept. For our analysis, we manually curate a set of realistic transformations (e.g., "snow" and "graffiti").

Note that this concept-transformation pipeline does not require *any* additional training or data annotation. Thus it can be directly applied to new datasets, as long as we have access to a pre-trained segmentation model for the concepts of interest.

## 4.2 Creating a suite of editing tasks

We now utilize the concept transformations described above to create a benchmark for evaluating model rewriting methods. Intuitively, these transformations can capture invariances that the model should ideally have—e.g., recognizing vehicles correctly even when they have wooden wheels. In practice however, model accuracy on one or more classes (e.g., "car", "scooter") may degrade under

these transformations. The goal of rewriting the model would thus be to fix these failure modes in a data-efficient manner. In this section, we evaluate our editing methodology—as well as the fine-tuning approaches discussed in Section 3—along this axis.

Concretely, we focus on vision classifiers—specifically, VGG [Simonyan and Zisserman, 2015] and ResNet [He et al., 2015] models trained on the ImageNet [Deng et al., 2009, Russakovsky et al., 2015] and Places-365 [Zhou et al., 2017] datasets (cf. Appendix A.2). Each test set is constructed using the concept-transformation pipeline discussed above, based on a chosen concept-style pair (say "wheel"-"wooden") [3]. It consists of $N$ exemplars (pairs of original and transformed images, $(x, x')$) that belong to a single (randomly-chosen) target class in the dataset. All other transformed images containing the concept, including those from classes other than the target one, are used for validation and testing (30-70 split). We create two variants of the test set: one using the same style image as the exemplars (i.e., same wooden texture) for the transformation; and another using held-out style images (i.e., other wooden textures).

To evaluate the impact of a method, we measure the change in model accuracy on the transformed examples (e.g., vehicles with "wooden wheel"s in Figure 2b). If the method is effective, then it should recover some of the incorrect predictions caused by the transformations. We only focus on the subset of examples $D$ that were correctly classified before the transformation, since we cannot expect to correct mistakes that do not stem from the transformation itself. Concretely, we measure the change in the number of mistakes made by the model on the transformed examples:

$$\% \text{ errors corrected} := \frac{N_{pre}(D) - N_{post}(D)}{N_{pre}(D)} \qquad (2)$$

where $N_{pre/post}(D)$ denotes the number of transformed examples misclassified by the model before and after the rewrite, respectively. Note that this metric can range from 100% when rewriting leads to perfect classification on the transformed examples, to even a negative value when the rewriting process causes more mistakes that it fixes.

In each case, we select the best hyperparameters—including the choice of the layer to modify—based on the validation set performance (cf. Appendix A.6.3). To quantify the effect of the modification on overall model behavior, we also measure the change in its (standard) test set performance. Since we are interested in rewrites that do not significantly hurt the overall model performance, we only consider hyperparameters that do not cause a large accuracy drop ($\leq 0.25\%$). We found that the exact accuracy threshold did not have significant impact on the results—see Appendix Figures 15-18 for a demonstration of the full accuracy-effectiveness trade-off.

## 4.3 The effectiveness of editing

Recall that a key desideratum of our prediction-rule edits is that they should *generalize*. That is, if we modify the way that our model treats a specific concept, we want this modification to apply to *every* occurrence of that concept. For instance, if we edit a model to enforce that "wooden wheels" should be treated the same as regular "wheels" in the context of "car" images, we want the model to do the same when encountering other vehicles with "wooden wheels". Thus, when analyzing performance in Figure 5, we consider inputs belonging to the class used to perform the edit separately.

**Editing.** We find that editing is able to consistently correct mistakes in a manner that *generalizes across classes*. That is, editing is able to reduce errors in non-target classes, often by more than 20 percentage points, even when performed using only three exemplars from the target class. Moreover, this improvement extends to transformations using different variants of the style (e.g., textures of "wood"), other than those present in exemplars used to perform the modification.

In Appendix B.1.3, we conduct ablation studies to get a better sense of the key algorithmic factors driving performance. Notably, we find that imposing the editing constraints (1) on the entirety of the image—as opposed to only focusing on key-value pairs that correspond to the concept of interest as proposed in Bau et al. [2020a]—leads to even better performance (cf. '-mask' in Figure 5). We hypothesize that this has a regularizing effect as it constrains the weights to preserve the original mapping between keys and values in regions that do not contain the concept.

---

[3]Note that some of these cases might not be suitable for editing, i.e., when the transformed concept is critical for recognizing the label of an input image (e.g., transforming concept "dog" in images of class "poodle"). We thus manually exclude such concept-class pairs from our analysis—cf. Appendix A.6.2.

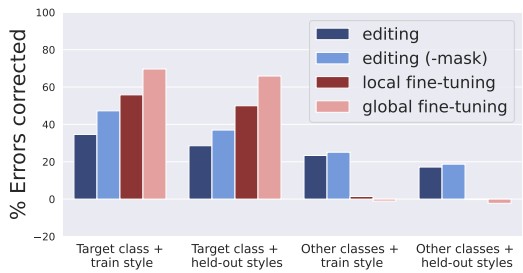 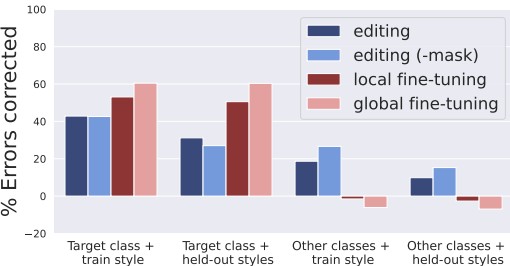

(a) ImageNet-trained VGG16 (concepts from COCO)  (b) Places-trained ResNet-18 (concepts from LVIS)

Figure 5: Editing vs. fine-tuning, averaged over concept-style pairs. We find that both methods (and their variants) are fairly successful at correcting errors on the target class (examples of which are used to perform the rewrite). This holds even when the transformation applied during testing is different from the one present in the train exemplars (e.g., a different texture of "wood"). However, crucially, only the improvements induced by editing generalize to other classes where the transformed concept is present. Fine-tuning fails in this setting—typically, causing more errors than it fixes. See Appendix Figures 10-13 for other experimental settings.

**Fine-tuning.** Our baseline is the canonical fine-tuning approach, i.e., directly minimizing the model loss on the new data (here the transformed images) with respect to their label. Similar to Section 3, we consider both the local and global variants of fine-tuning. We find that while these approaches are able to correct model errors on transformed inputs from the class used to perform the modification, they typically *decrease* performance on *other* classes—i.e., they cause more errors than they fix. Moreover, even when we allow a larger drop in the model's accuracy, or use more training exemplars, their performance often becomes *worse* such inputs (Appendix Figures 15-18).

We present examples of errors corrected (or not) by editing and fine-tuning in Appendix Figure 14, and provide a per-concept/style break down in Appendix Figures 19 and 20.

## 5 Beyond editing: Probing model behavior via counterfactuals

In the previous section, we developed a scalable pipeline for creating concept-level transformations which we used to evaluate model rewriting methods. Here, we put forth another related use-case of that pipeline: debugging models to identify their (learned) prediction rules.

In particular, observe that the resulting transformed inputs (cf. Figure 4) can be viewed as *counterfactuals*—a primitive commonly used in causal inference [Pearl, 2010] and interpretability [Goyal et al., 2019a,b, Bau et al., 2020b]. Counterfactuals can be used to identify the features that a model uses to make its prediction on a given input. For example, to understand whether the model relies on the presence of a "wheel" to recognize a "car" image, we can evaluate how the its prediction changes when just the wheel in the image is transformed. Based on this view, we now repurpose our concept transformations from Section 4 to discover a given classifier's prediction rules.

**The effect of specific concepts.** As a point of start, we study how sensitive the model's prediction is to a given concept in terms of the accuracy drop caused by transformation of said concept. For instance, in Figure 6a, we find that the accuracy of a VGG16 ImageNet classifier drops by 25% on images of "croquet ball" when "grass" is transformed, whereas its accuracy on "collie" does not change. In line with previous studies [Zhang et al., 2007, Ribeiro et al., 2016, Rosenfeld et al., 2018, Barbu et al., 2019, Xiao et al., 2020], we also find that background concepts, such as "grass", "sea" and "sand", have a large effect on model performance. We contrast this measure of influence across concepts for a single model (Appendix Figure 27), and across architectures for a single concept (Appendix B.2). Finally, we examine the effect of each concept stylization in Figure 6b.

**Per-class prediction rules.** If instead we restrict our attention to a single class, we can pinpoint the set of concepts that the model relies on to detect this class. It turns out that aside from the main image object, ImageNet classifiers also heavily depend on co-occurring objects [Stock and Cisse,

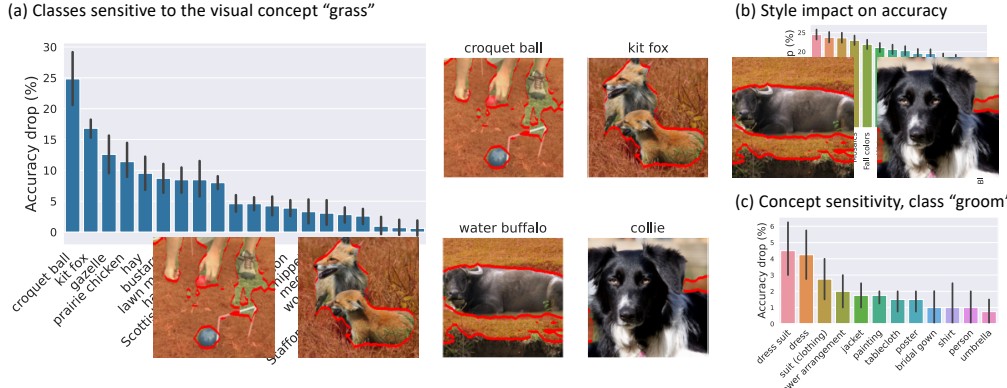

Figure 6: Model sensitivities diagnosed using our pipeline in a VGG16 ImageNet classifier. (a) Classes for which the model relies on "grass": e.g., a "croquet ball" is not accurately recognized if "grass" is transformed, while "collie"'s are not affected. (b) Applying different transformations to visual concepts reduces accuracy by varying amounts. (c) Transformations that hamper accuracy can highlight prediction rules: e.g., the class "groom" is sensitive to the presence of "dress."

2018, Tsipras et al., 2020, Beyer et al., 2020] in the image—e.g., "dress" for the class "groom", "person" for the class "tench" (*sic*), and "road" for the class "race car" (cf. Figure 6c and Appendix Figure 29). We can also examine which styles hurt performance the most—e.g., we find that making "plants" "floral" hurts accuracy on the class "damselfly" 15% more than making them "snowy".

Overall, this pipeline provides a scalable path for model designers to analyze the invariances (and sensitivities) of their models with respect to various natural transformations of interest. We thus believe that this primitive holds promise for future interpretability and robustness studies.

# 6 Related work

**High-level concepts in latent representations.** There has been increasing interest in explaining the inner workings of deep classifiers through high-level concepts: e.g., by identifying individual neurons [Erhan et al., 2009, Zeiler and Fergus, 2014, Olah et al., 2017, Bau et al., 2017, Engstrom et al., 2019a] or activation vectors [Kim et al., 2018, Zhou et al., 2018, Chen et al., 2020] that correspond to human-understandable features. The effect of these features on model predictions can be analyzed by either inspecting the downstream weights of the model [Olah et al., 2018, Wong et al., 2021] or through counterfactual tests: e.g., via synthetic data [Goyal et al., 2019a]; by swapping features between individual images [Goyal et al., 2019b]; or by silencing sets of neurons [Bau et al., 2020b]. A parallel line of work aims to learn models that operate on data representations that explicitly encode high-level concepts: either by learning to predict a set of attributes [Lampert et al., 2009, Koh et al., 2020a] or by learning to segment inputs [Losch et al., 2019]. In our work, we identify concepts by manually selecting the relevant pixels in a handful of images and measure the impact of manipulating these features on model performance on a new test set.

**Model interventions.** Direct manipulations of latent representations inside generative models have been used to create human-understandable changes in synthesized images [Bau et al., 2019, Jahanian et al., 2019, Goetschalckx et al., 2019, Shen et al., 2020, Härkönen et al., 2020, Wu et al., 2020]. Our work is inspired by that line of work as well as a recent finding that parameters of a generative model can be directly changed to alter generalized behavior [Bau et al., 2020a]. Unlike previous work, we edit classification models, changing rules that govern predictions rather than image synthesis. Concurrently with our work, there has been a series of methods proposed for editing factual knowledge in language models [Mitchell et al., 2021, De Cao et al., 2021, Dai et al., 2021].

**Ignoring spurious features.** Prior work on preventing models from relying on spurious correlations is based on constraining model predictions to satisfy certain invariances. Examples include: training on counterfactuals (either by adding or removing objects from scenes [Shetty et al., 2018, 2019, Agarwal et al., 2020] or having human annotators edit text input [Kaushik et al., 2019]),

learning representations that are simultaneously optimal across domains [Arjovsky et al., 2019], ensuring comparable performance across subpopulations [Sagawa et al., 2020], or enforcing consistency across inputs that depict the same entity [Heinze-Deml and Meinshausen, 2017]. In this work, we focus on a setting where the model designer is aware of undesirable correlations learned by the model and we provide the tools to rewrite them directly.

**Model robustness.** A long line of work has been devoted to discovering and correcting failure modes of models. These studies focus on simulating variations in testing conditions that can arise during deployment, including: adversarial or natural input corruptions [Szegedy et al., 2014, Fawzi and Frossard, 2015, Fawzi et al., 2016, Engstrom et al., 2019b, Ford et al., 2019, Hendrycks and Dietterich, 2019, Kang et al., 2019], changes in the data collection process [Saenko et al., 2010, Torralba and Efros, 2011, Khosla et al., 2012, Tommasi and Tuytelaars, 2014, Recht et al., 2019], or variations in the data subpopulations present [Beery et al., 2018, Oren et al., 2019, Sagawa et al., 2020, Santurkar et al., 2021, Koh et al., 2020b]. Typical approaches for improving robustness in these contexts include robust optimization [Madry et al., 2018, Yin et al., 2019, Sagawa et al., 2020] and data augmentation schemes [Lopes et al., 2019, Hendrycks et al., 2019, Zhang et al., 2021]. Our rule-discovery and editing pipelines can be viewed as complementary to this work as they allows us to preemptively adjust the model's prediction rules in anticipation of deployment conditions.

**Domain adaptation.** The goal of domain adaptation is to adapt a model to a specific deployment environment using (potentially unlabeled) samples from it. This is typically achieved by either fine-tuning the model on the new domain [Donahue et al., 2014, Sharif Razavian et al., 2014, Kumar et al., 2020], learning the correspondence between the source and target domain, often in a latent representation space [Ben-David et al., 2007, Saenko et al., 2010, Ganin and Lempitsky, 2015, Courty et al., 2016, Gong et al., 2016], or updating the model's batch normalization statistics [Li et al., 2016, Burns and Steinhardt, 2021]. These approaches all require a non-trivial amount of data from the target domain. The question of adaptation from a handful of samples has been explored [Motiian et al., 2017], but in a setting that requires samples across all target classes. In contrast, our method allows for generalization to new (potentially unknown) classes with even a single example.

# 7 Conclusion

We developed a general toolkit for performing targeted post hoc modifications to vision classifiers. Crucially, instead of specifying the desired behavior *implicitly* via the training data, our method allows users to *directly* edit the model's prediction rules. By doing so, our approach makes it easier for users to encode their prior knowledge and preferences during the model debugging process. Additionally, a key benefit of this technique is that it fundamentally changes how the model processes a given concept—thus making it possible to edit its behavior beyond the specific class(es) used for editing. Finally, our edits do not require any additional data collection: they can be guided by as few as a single (synthetically-created) exemplar. We believe that this primitive opens up new avenues to interact with and correct our models before or during deployment.

**Limitations and broader impact.**

Even though our methodology provides a general tool for model editing, performing such edits does require manual intervention and domain expertise. After all, the choice of what concept to edit—and its implications on the robustness of the model—lies with the model designer. For instance, in the vehicles-on-snow example, our objective was to have the model recognize any vehicle on snow the same way it would on a regular road—e.g., to adapt a system to different weather conditions. However, if our dataset contains classes for which the presence snow is absolutely essential for recognition, this might not be an appropriate edit to perform.

Moreover, direct model editing is a departure from the standard way in which models are trained, and may have broader implications. While we have shown how it can be used to cause beneficial changes in pre-trained models, direct control of prediction rules could also make it easier for adversaries to introduce vulnerabilities into the model (e.g., by manipulating model behavior on a specific population demographic). Overall, direct model editing makes it clearer than ever that our models are a reflection of the goals and biases of we who create them—not only through the training tasks we choose, but now also through the rules that we rewrite.

## Acknowledgments and Disclosure of Funding

We thank the anonymous reviewers for their helpful comments and feedback.

Work supported in part by the NSF grants CCF-1553428 and CNS-1815221, the DARPA SAIL-ON HR0011-20-C-0022 grant, Open Philanthropy, a Google PhD fellowship, and a Facebook PhD fellowship. This material is based upon work supported by the Defense Advanced Research Projects Agency (DARPA) under Contract No. HR001120C0015.

Research was sponsored by the United States Air Force Research Laboratory and the United States Air Force Artificial Intelligence Accelerator and was accomplished under Cooperative Agreement Number FA8750-19-2-1000. The views and conclusions contained in this document are those of the authors and should not be interpreted as representing the official policies, either expressed or implied, of the United States Air Force or the U.S. Government. The U.S. Government is authorized to reproduce and distribute reprints for Government purposes notwithstanding any copyright notation herein.

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
