# A  Experimental setup

## A.1  Datasets

For the bulk of our experimental analysis (Sections 4 and 5) we use the ImageNet-1k [Deng et al., 2009, Russakovsky et al., 2015] and Places-365 [Zhou et al., 2017] datasets which contain images from 1,000 and 365 categories respectively. In particular, both prediction-rule discovery and editing are performed on (or using) samples from the standard test sets to avoid overlap with the training data used to develop the models.

**Licenses.**  Both datasets were collected by scraping online image hosting engines and, thus, the images themselves belong to their individuals who uploaded them. Nevertheless, as per the terms of agreement for each dataset [4] [5], these images can be used for non-commercial research purposes.

## A.2  Models

Here, we describe the exact architecture and training process for each model we use. For most of our analysis, we utilize two canonical, yet relatively diverse model architectures for our study: namely, VGG [Simonyan and Zisserman, 2015] and ResNet [He et al., 2016]. We use the standard PyTorch implementation [6] and train the models from scratch on the ImageNet and Places365 datasets. The accuracy of each model on the corresponding test set is provided in Table 1.

**ImageNet classifiers.**  We study: (i) a VGG16 variant with batch normalization and (ii) a ResNet-50. Both models are trained using standard hyperparameters: SGD for 90 epochs with an initial learning rate of 0.1 that drops by a factor of 10 every 30 epochs. We use a momentum of 0.9, a weight decay of $10^{-4}$ and a batch size of 256 for the VGG16 and 512 for the ResNet-50.

**Places365 classifiers.**  We study: (i) a VGG16 and (ii) a ResNet-18. Both models are trained for 131072 iterations using SGD with a single-cycle learning rate schedule peaking at 2e-2 and descending to 0 at the end of training. We use a momentum 0.9, a weight decay 5e-4 and a batch size of 256 for both models.

**CLIP.**  For the typographic attacks of Section 3, we use the ResNet-50 models trained via CLIP [Radford et al., 2021], as provided in the original model repository.[7]

|  | Test Accuracy (%) | |
|---|---|---|
| Architecture \ Dataset | ImageNet | Places |
| VGG | 73.70 | 54.02 |
| ResNet | 75.77 | 54.24 |
| CLIP-ResNet | 59.84 | - |

Table 1: Accuracy of each model architecture on the datasets used in our analysis.

## A.3  Compute

Our experiments were performed on our internal cluster, comprised mainly of NVIDIA 1080Ti GTX GPUs. For the rule-discovery pipeline, we only need to evaluate models on images where the corresponding object is present, which allows us to perform the evaluation on a single style in less than 8 hours on a single GPU (amortized over concepts and classes). For the model rewriting process, each instance of editing or fine-tuning takes a little more than a minute, since it only operates on a section of the model and using a handful of training examples.

---

[4] http://places2.csail.mit.edu/download.html
[5] https://www.image-net.org/about.php
[6] https://pytorch.org/vision/stable/models.html
[7] https://github.com/openai/CLIP

### A.4 Model rewriting

Here, we describe the training setup of our model editing process, as well as the fine-tuning baseline. Recall that these rewrites are performed with respect to a single concept-style pair.

**Layers.** We consider a layer to be a block of convolution-BatchNorm-ReLU, similar to Bau et al. [2020a] and rewrite the weights of the convolution. For ResNets (which were not previously studied), we must also account for skip connections. In particular, note that the effect of a rewrite to a layer inside any residual block will be attenuated (or canceled) by the skip connection. To avoid this, we only rewrite the final layer within each residual block—i.e., focus on the convolution-BatchNorm-ReLU right before a skip connection, and include the skip connection in the output of the layer. Unless otherwise specified, we perform rewrites to layers $[8, 10, 11, 12]$ for VGG models, $[4, 6, 7]$ for ResNet-18, and $[8, 10, 14]$ for ResNet-50 models. We tried earlier layers in our initial experiments, but found that both methods perform worse.

#### A.4.1 Editing

We use the ADAM optimizer with a fixed learning rate to perform the optimization in (1). We grid over different learning rate-number of step pairs: $[(10^{-3}, 10000), (10^{-4}, 20000), (10^{-5}, 40000), (10^{-6}, 80000), (10^{-7}, 80000)]$. The second order statistics (cf. Section 2) are computed based on the keys for the entire test set.

#### A.4.2 Fine-tuning

When fine-tuning a single layer (local fine-tuning), we optimize the weights of the convolution of that particular layer. Instead, when we fine-tune a suffix of the model (global fine-tuning), we optimize all the trainable parameters including and after the chosen layer. In both cases, we use SGD, griding over different learning rate-number of step pairs: $[(10^{-2}, 500), (10^{-3}, 500), (10^{-4}, 500), (10^{-5}, 800), (10^{-6}, 800)]$. We verified that in all cases the optimal performance of the method was achieved for hyperparameters strictly within that range and thus performing more steps would not provide any benefits.

### A.5 Real-world test cases

In Section 3 we study two real-world applications of our model rewriting methodology. Below, we outline the data-collection process for each case as well as the hyperparameters used.

**Vehicles on snow.** We manually chose a subset of Imagenet classes that frequently contain "roads", identified using our prediction-rule discovery pipeline in Section 4. In particular, we focus on the classes: "racing car", "army tank", "fire truck", "car wheel", "traffic light", "school bus", and "motor scooter". For each of these classes, we searched Flickr[8] using the query "<class name> on snow" and manually selected the images that clearly depicted the class and actually contained snowy roads. We were able to collect around 20 pictures for each class with the exception of "traffic light" where we only found 9.

**Typographic attacks.** We picked six household objects corresponding to ImageNet classes, namely: "teapot", "mug", "flower pot", "toilet tissue", "vase", and "wine bottle". We used a smartphone camera to photograph each of these objects against a plain background. Then, we repeated this process but after affixing a piece of paper with the text "iPod" handwritten on it, as well as when affixing a blank piece of paper—see Figure 7.

**Hyperparameters.** Since our manually collected test sets are rather small, we decided to avoid tuning hyperparameters on them as this would require holding out a non-trivial number of data points. Instead, we inspected the results of the large-scale synthetic evaluation and manually picked values that performed consistently well which we list in Table 2. Nevertheless, we found that the results would be quite similar if we tuned hyperparameters directly on these test sets.

---

[8] https://www.flickr.com/

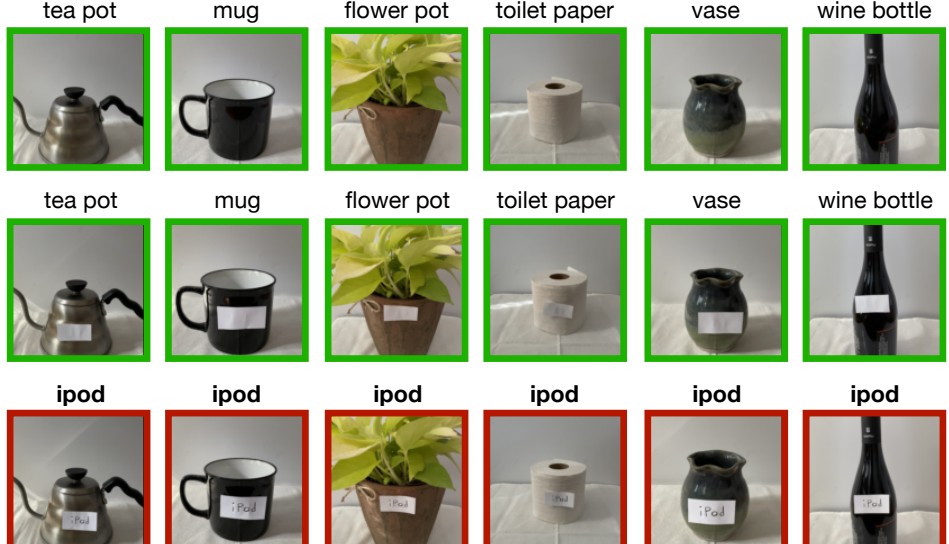

Figure 7: Typographic attacks on CLIP: We reproduce the results of Goh et al. [2021] by taking photographs of household objects with a paper containing handwritten text "iPod" attached to them (third row). We see that these attacks consistently fool the zero-shot CLIP classifier (ResNet-50)— compare the predictions (shown in the title) for the first and third row. In contrast, if we instead use a blank piece of paper (second row), the model predicts correctly.

| Model | Method | # steps | Step size | Layer | Mask |
|---|---|---|---|---|---|
| VGG16 | Editing | 20,000 | 1e-4 | 12 (last) | No |
| | Fine-tuning | 400 | 1e-4 | 12 (last) | N/A |
| CLIP-ResNet-50 | Editing | 20,000 | 1e-4 | 14 (last) | No |
| | Fine-tuning | 800 | 1e-5 | 14 (last) | N/A |

Table 2: Hyperparameters chosen for evaluating on the real-world test cases.

## A.6 Synthetic evaluation

We now describe the details of our evaluation methodology, namely, how we transform inputs, how we chose which concept-style pairs to use for testing and how we chose the hyperparameters for each method.

### A.6.1 Creating concept-level transformations

Recall that our pipeline for transforming concepts consists of two steps: concept detection and concept transformation (Section 4). We describe each step below and provide examples in Figure 8.

We detect concepts using pre-trained object detectors trained on MS-COCO [Lin et al., 2014] and LVIS [Gupta et al., 2019]. For MS-COCO, we use a model with a ResNet-101 backbone[9] which is trained on COCO-Stuff[10] annotations and can detect 182 concepts [Chen et al., 2017]. For LVIS, we use a pre-trained model from the Detectron [Girshick et al., 2018] model zoo[11], which can detect 1230 classes. We only consider a prediction as valid for a specific pixel if the model's predicted probability is at least 0.80 for the COCO-based model and 0.15 for the LVIS-based model (chosen based on manual inspection). Moreover, we treat a concept as present in a specific image if it present in at least 100 pixels (image size is 224×224 for ImageNet and 256×256 for Places).

---

[9]https://github.com/kazuto1011/deeplab-pytorch
[10]https://github.com/nightrome/cocostuff
[11]https://github.com/facebookresearch/detectron2/blob/master/MODEL_ZOO.md

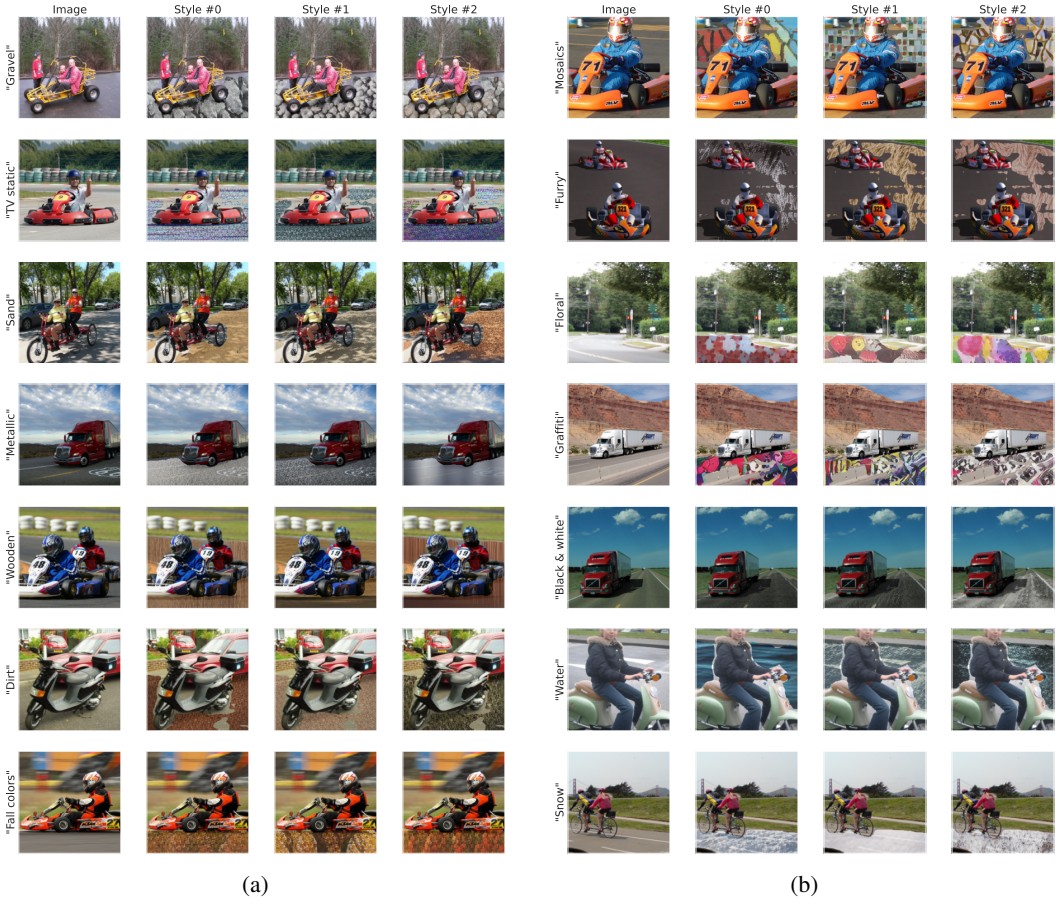

(a)                                                                                    (b)

Figure 8: Illustration of concept-level transformations in ImageNet: We transform the concept "road" in images belonging to various classes via style transfer. Each row (within (a) and (b)) depicts the stylization of a single image with respect to the style described in the label (e.g., "gravel"). We collect three examples per style, which are then split across training and testing.

In order to transform concepts, we utilize the fast style transfer methodology of Ghiasi et al. [2017] using their pre-trained model[12]. This allows us to quickly apply the same style to a large number of images which is ideal for our use-case. Specifically, we manually choose 14 styles (illustrated in Figure 8) and choose 3 images for each. This allows us to perform the concept-level transformation in several ways and evaluate how sensitive our model is to the exact style used.

All the pre-trained models used are open-sourced and freely available for non-commercial research.

### A.6.2  Selecting concept-style pairs

**Concept selection.** Recall that our transformation pipeline from Section 4 identifies concepts which, when transformed in a certain manner hurts model accuracy on one or more classes. We first filter these concepts (automatically) to identify ones that are particularly salient in the model's prediction-making process. In particular, we focus on concepts which simultaneously: (a) affect at least 3 classes; (b) are present in at least 20% percent of the test images of each class; and (c) cause a drop of at least 15% among these images. This selection results in a test bed where we can meaningfully observe differences in performance between approaches.

At the same time, we need to also ensure that the rewriting task we are solving is meaningful. For instance, if we replace all instances of "dog" with a stylized version, then distinguishing between a "terrier" and a "poodle" can become challenging (or even impossible). Moreover, we cannot expect

---

[12]https://tfhub.dev/google/magenta/arbitrary-image-stylization-v1-256/2

the performance of the model to improve on other dog breeds if we modify it to treat a stylized dog as a "terrier". To eliminate such test cases, we manually filter the concept-class pairs flagged by our prediction-rule discovery pipeline. In particular, we removed those where the detected concept overlapped significantly with the class object itself. In other words, if the concept detected is essential for correctly recognizing the class of the image, we exclude it from our analysis. Typical examples of excluded concept-class pairs on ImageNet include broad animal categories (e.g., "bird" or "dog") for classes corresponding to specific breeds (e.g., "parrot") or the concept "person" which overlaps with classes corresponding to articles of clothing (e.g., "suit").

**Style selection.** We consider a subset of 8 styles for our analysis: "black and white", "floral", "fall colors", "furry", "graffiti", "gravel", "snow" and "wooden". While performing editing with respect to a single concept-style pair—say "wheel"-"wooden"—we randomly select one wooden texture to create train exemplars and hold out the other two for testing (described as held-out styles in the figures).

### A.6.3 Hyperparameter selection

As discussed in Appendix A.4, for a particular concept-style pair, we grid over different hyperparameters pertaining to the rewrite (via editing or fine-tuning)—in particular the layer that is modified, as well as training parameters such as the learning rate. For our evaluation, we then choose a single set of hyperparameters (per concept-style pair). At a high level, our objective is to find hyperparameters that improve model performance on transformed examples, while also ensuring that the test accuracy of the model does not drop below a certain threshold. To this end, we create a validation set per concept-style pair with 30% of the examples containing this concept (and transformed using the same style as the train exemplars). We then use the performance on that subset (2) to choose the best set of hyperparameters. If all of the hyperparameters considered cause accuracy to drop below the specified threshold, we choose to not perform the edit at all. We then report the performance of the method on the test set (the other 70% of samples containing this concept).

### A.7 Data ethics

Since we manually collected all the data necessary for our analysis in Section 3, we were able to filter them for offensive content. Moreover, we made sure to only collect images that are available under a Creative Commons license (hence allowing non-commercial use with proper attribution).

For the rest of our analysis, we relied on publicly available datasets that are commonly used for image classification. Unfortunately, due to their scale, these datasets have not been thoroughly filtered for offensive content or identifiable information. In fact, improving these datasets along this axis is an active area of work[13]. Nevertheless, since our research did not involve redistributing these datasets or presenting them to human annotators, we did not perceive any additional risks that would result from our work.

---

[13]https://www.image-net.org/update-mar-11-2021.php

# B  Additional experiments

## B.1  Fine-grained model behavior on typographic attacks

In Figure 9, we take a closer look at how effective different rewriting methods (with one train exemplar) are in mitigating typographic attacks. We find that:

- *Local-finetuning:* Corrects only a subset of the errors.
- *Global-finetuning:* Corrects most errors on the attacked images. However, on the flip side it: (i) causes the model to spuriously associate other images with the target class ("teapot") used to perform fine-tuning and (ii) significantly reduces model accuracy on clean images of the class "iPod".
- *Editing:* Corrects all errors without substantially hurting model accuracy on clean images.

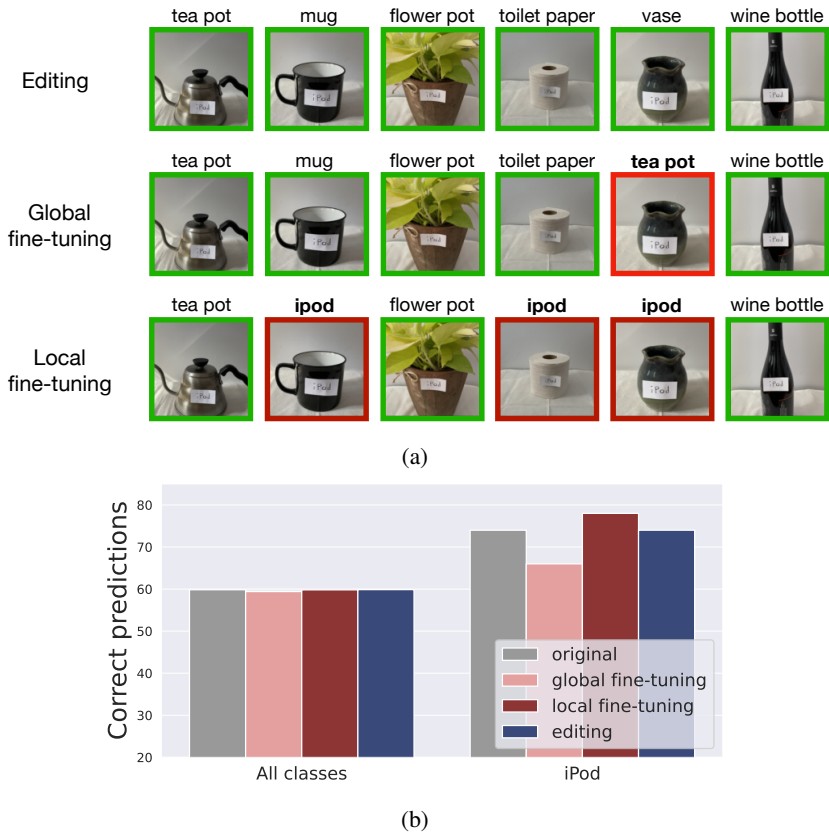

Figure 9: Effectiveness of different modification procedures in preventing typographic attacks. (a) Model predictions after the rewrite—local fine-tuning often fails to prevent such attacks, while global fine-tuning results in the model associating "iPod" with the target class used for fine-tuning ("teapot"). (b) Accuracy on the original test set and specifically on clean samples from class "ipod" before and after the rewrite. While global fine-tuning is fairly effective at mitigating typographic attacks, it disproportionately reduces model accuracy on clean images from the "iPod" class.

### B.1.1 The effectiveness of editing

In Figures 10- 13, we compare the generalization performance of editing and fine-tuning (and their variants)—for different datasets (ImageNet and Places), architectures (VGG16 and ResNets) and number of exemplars (3 and 10). In performing these evaluations, we only consider hyperparameters (for each concept-style pair) that do not drop the overall (test set) accuracy of the model by over 0.25%. The complete accuracy-performance trade-offs of editing and fine-tuning (and their variants) are illustrated in Appendix Figures 15-18.

We observe that both methods successfully generalize to held-out samples from the target class (used to perform the modification)—even when the transformation is performed using held-out styles. However, while the performance improvements of editing also extend to other classes containing the same concept, this does not seem to be the case for fine-tuning. These trends hold even when we use more exemplars to perform the modification. In Appendix Figure 14, we illustrate sample error corrections (and failures to do so) due to editing and fine-tuning.

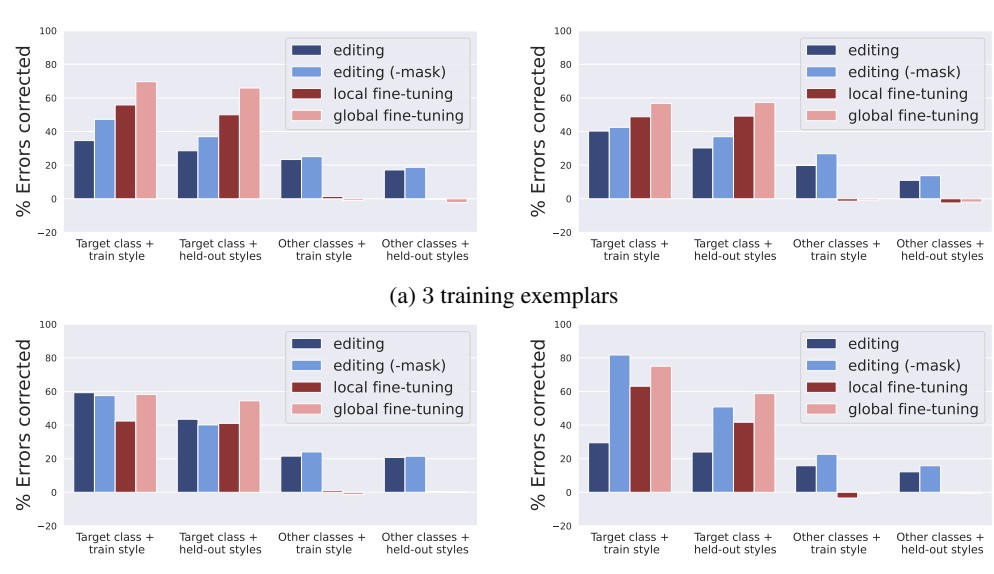

(a) 3 training exemplars

(b) 10 training exemplars

Figure 10: Editing vs. fine-tuning: average number of misclassifications corrected by the method when applied to an ImageNet-trained VGG16 classifier. Here, the average is computed over different concept-transformation pairs—with concepts derived from instance segmentation modules trained on MS-COCO (*left*) and LVIS (*right*); and transformations described in Appendix A.4.1. For both editing and fine-tuning, the overall drop in model accuracy is less than 0.25%.

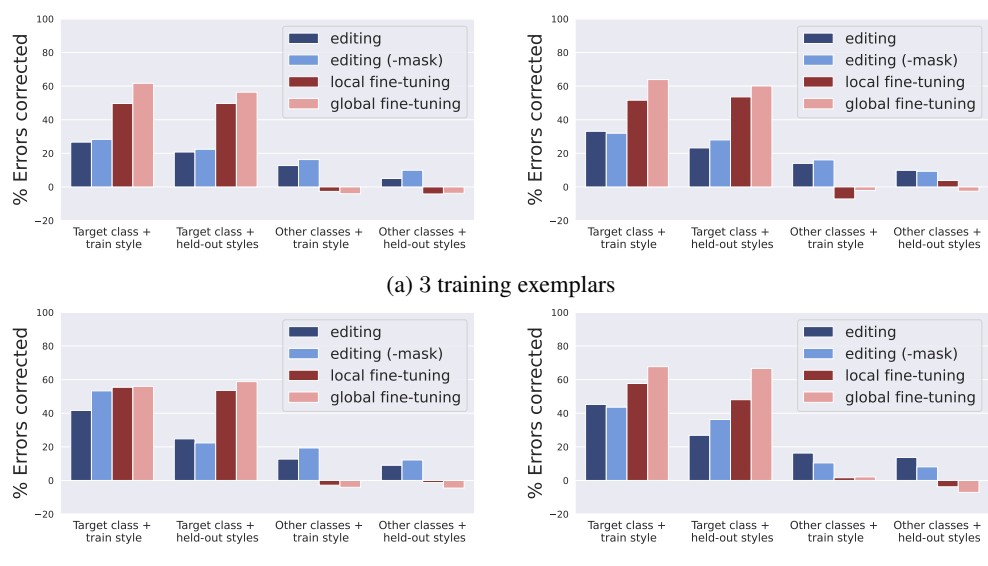

(a) 3 training exemplars

(b) 10 training exemplars

Figure 11: Repeating the analysis in Appendix Fig. 10 on an ImageNet-trained ResNet-50 classifier.

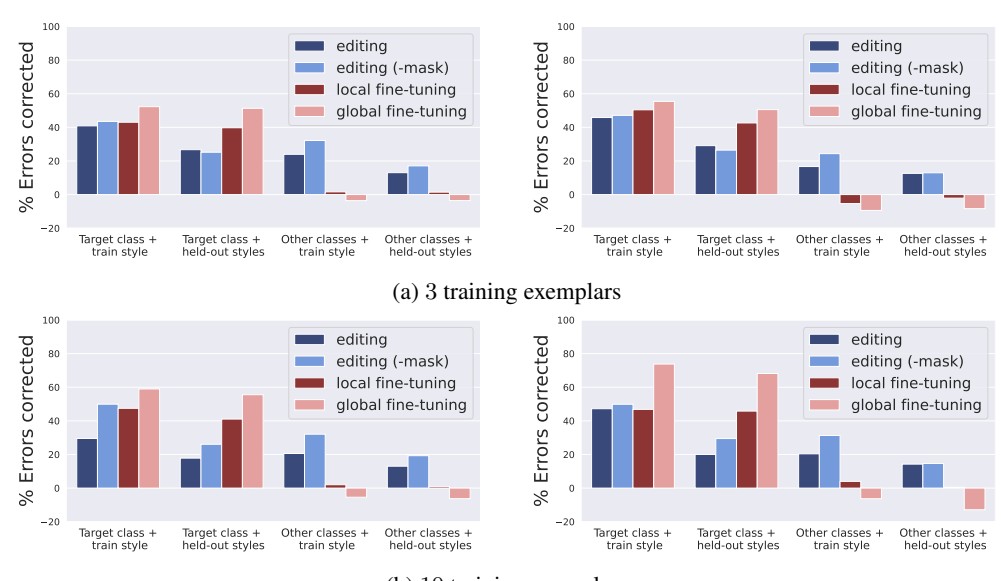

(a) 3 training exemplars

(b) 10 training exemplars

Figure 12: Repeating the analysis in Appendix Fig. 10 on an Places365-trained VGG16 classifier.

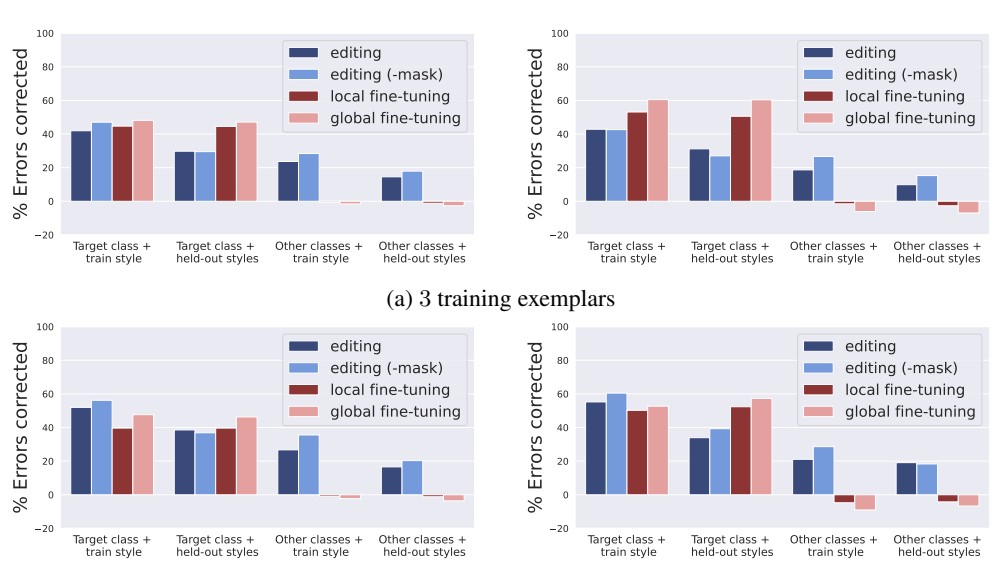

(a) 3 training exemplars

(b) 10 training exemplars

Figure 13: Repeating the analysis in Appendix Fig. 10 on an Places365-trained ResNet-18 classifier.

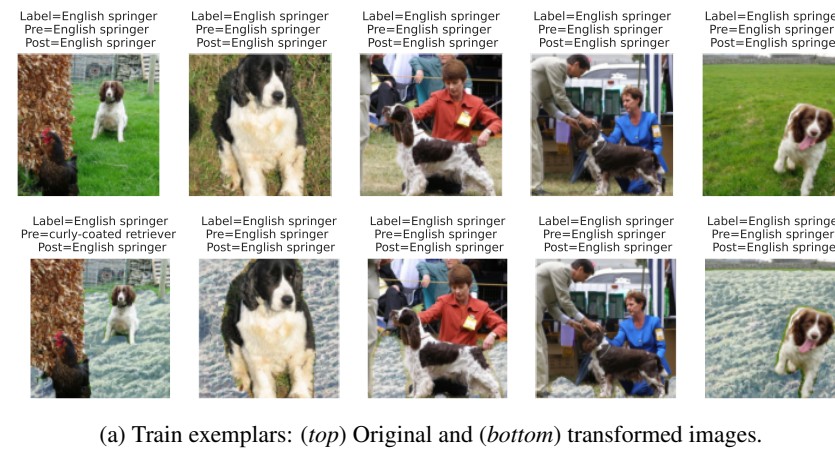

(a) Train exemplars: (*top*) Original and (*bottom*) transformed images.

## Method: Fine-tuning

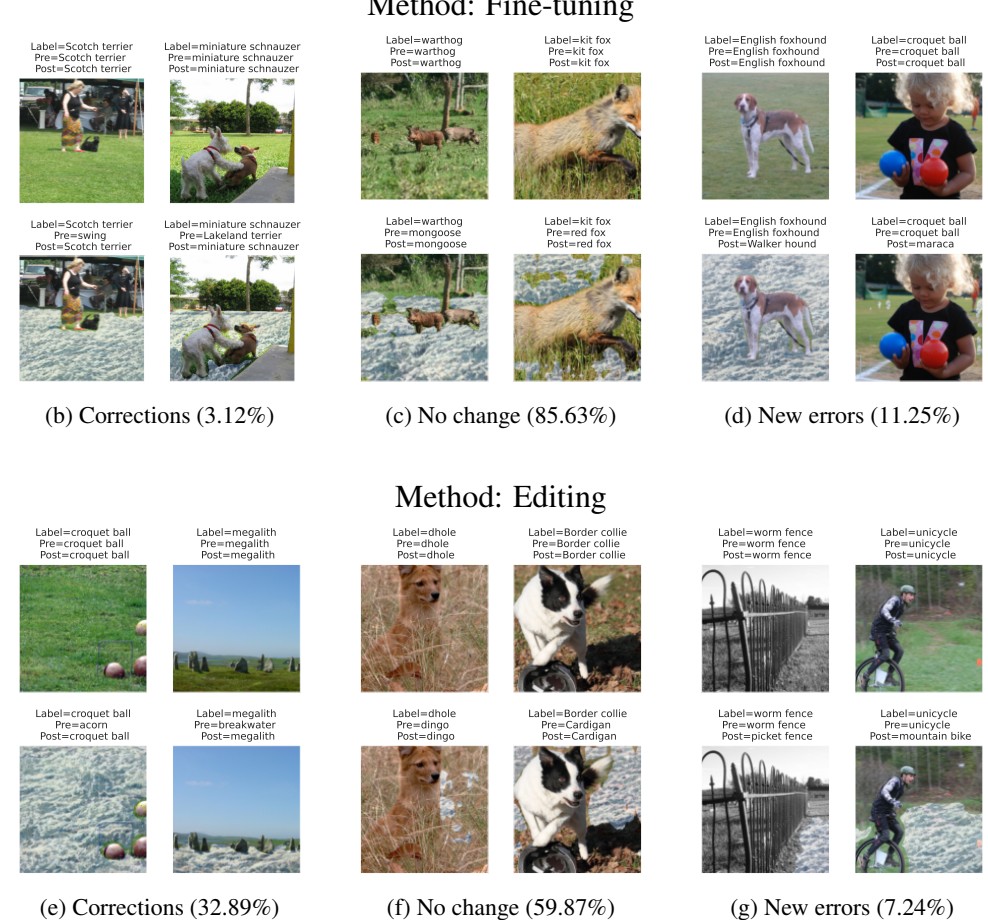

(b) Corrections (3.12%)  (c) No change (85.63%)  (d) New errors (11.25%)

## Method: Editing

(e) Corrections (32.89%)  (f) No change (59.87%)  (g) New errors (7.24%)

Figure 14: Examples of errors (not) corrected by the rewrite: Here, the goal is to improve the accuracy of a VGG16 model when "grass" in ImageNet images is replaced with "snow". The true label, as well as model predictions pre/post-edit for each image are in the title. (a) Train exemplars for editing and fine-tuning. Test set examples where fine-tuning and editing correct the model error on the transformed example (b/e), do not cause any change (c/f) and induce an new error (d/g). The number in parenthesis indicates the fraction of the test set that falls into each of these subsets.

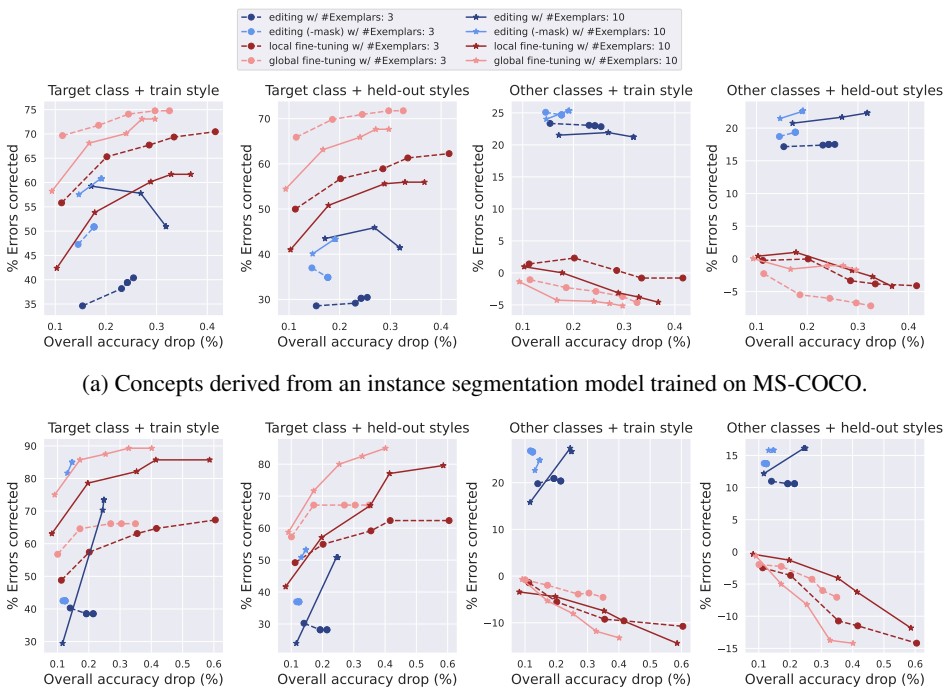

(a) Concepts derived from an instance segmentation model trained on MS-COCO.

Figure 15: Performance vs. drop in overall test set accuracy: Here, we visualize average number of misclassifications corrected by editing and fine-tuning when applied to an ImageNet-trained VGG16 classifier—where the average is computed over different concept-transformation pairs.

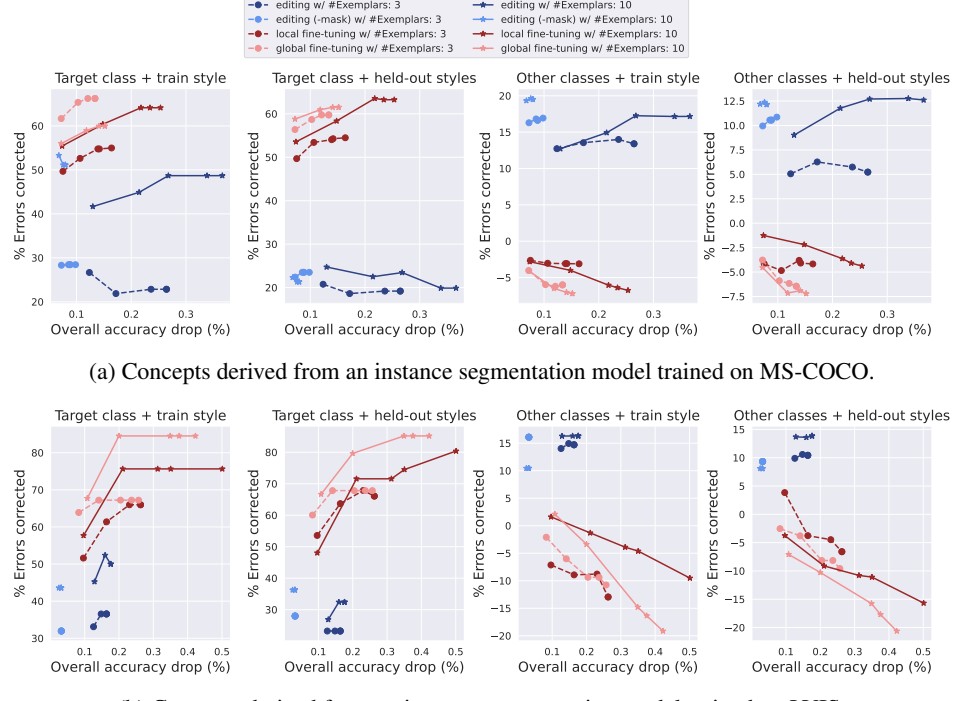

(a) Concepts derived from an instance segmentation model trained on MS-COCO.

(b) Concepts derived from an instance segmentation model trained on LVIS.

Figure 16: Repeating the analysis in Appendix Fig. 15 on an ImageNet-trained ResNet-50 classifier.

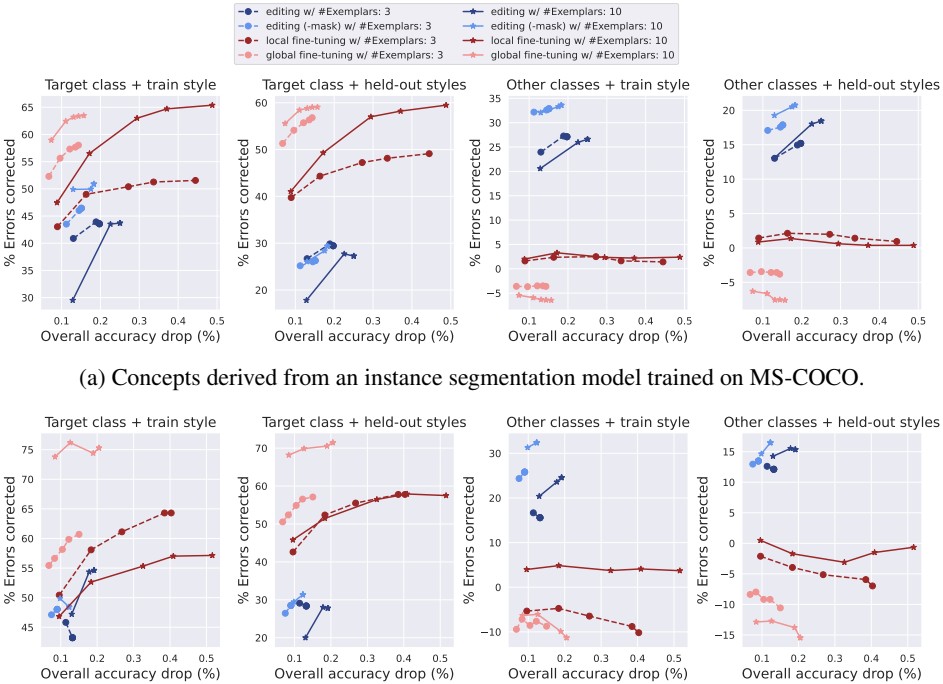

(a) Concepts derived from an instance segmentation model trained on MS-COCO.

(b) Concepts derived from an instance segmentation model trained on LVIS.

Figure 17: Repeating the analysis in Appendix Fig. 15 on an Places365-trained VGG16 classifier.

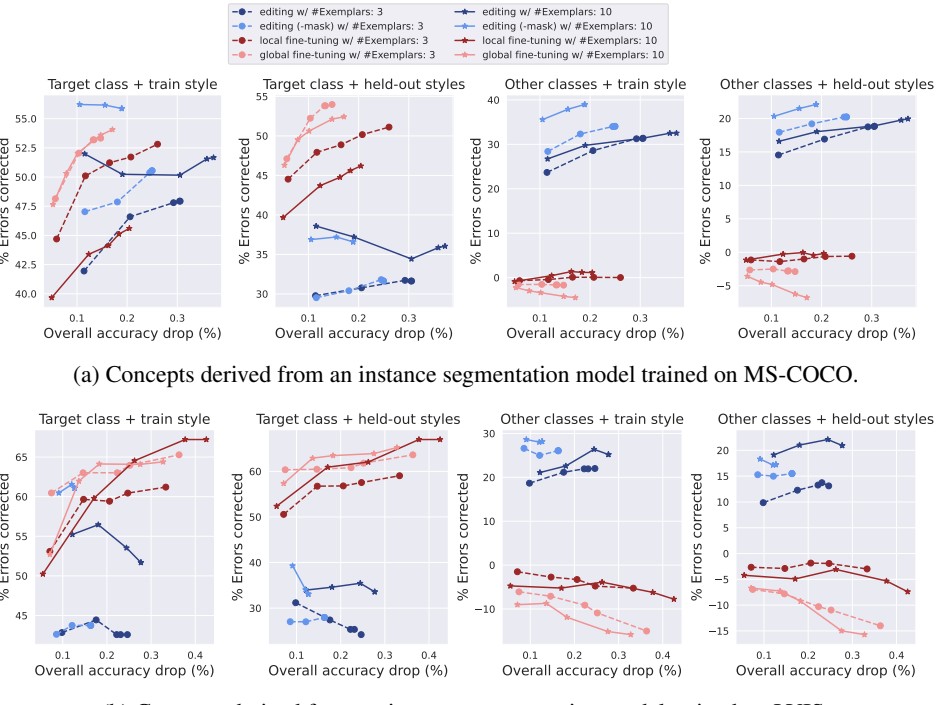

(a) Concepts derived from an instance segmentation model trained on MS-COCO.

(b) Concepts derived from an instance segmentation model trained on LVIS.

Figure 18: Repeating the analysis in Appendix Fig. 15 on an Places365-trained ResNet-18 classifier.

### B.1.2  A fine-grained look at performance improvements

In Appendix Figures 19 and 20 we take a closer look at the performance improvements caused by editing (-mask) and (local) fine-tuning (with 10 exemplars) on an ImageNet-trained VGG16 classifier. In particular, we break down the improvements on test examples from *non-target* classes with the same transformation as training, per-concept and per-style respectively. As before, we only consider hyperparameters that lead to an overall accuracy drop of less than $0.25\%$.

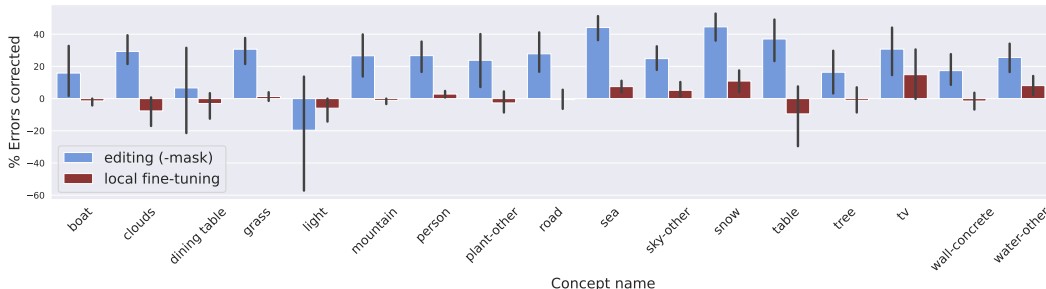

(a) Concepts derived from an instance segmentation model trained on MS-COCO.

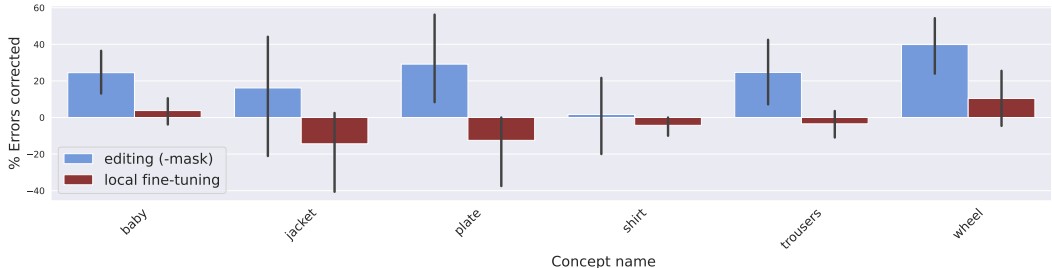

(b) Concepts derived from an instance segmentation model trained on LVIS.

Figure 19: Performance of editing and fine-tuning on test examples from non-target classes containing a given concept, averaged across transformations (cf. Appendix A.4.1).

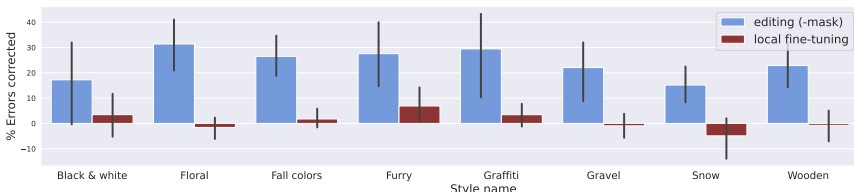

(a) Concepts derived from an instance segmentation model trained on MS-COCO.

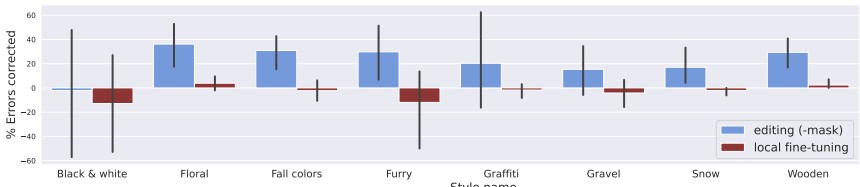

(b) Concepts derived from an instance segmentation model trained on LVIS.

Figure 20: Performance of editing and fine-tuning on test examples from non-target classes transformed using a given style, averaged across concepts.

### B.1.3 Ablations

In order to get a better understanding of the core factors that affect performance in this setting, we conduct a set of ablation studies. Note that we can readily perform these ablations as, in contrast to the setting of Bau et al. [2020a] we have access to a quantitative performance metric that does not rely on human evaluation.

**Layer.** We compare both editing and local fine-tuning when they are applied to different layers of the model in Appendix Figure 21. For editing, we find a consistent increase in performance—on examples from both the target and other classes—as we edit deeper into the model. For (local) fine-tuning, a similar trend is observed with regards to performance on the target class, with the second last layer being optimal overall. However, at the same time, the fine-tuned model's performance on examples from other classes containing the concept seems to get worse.

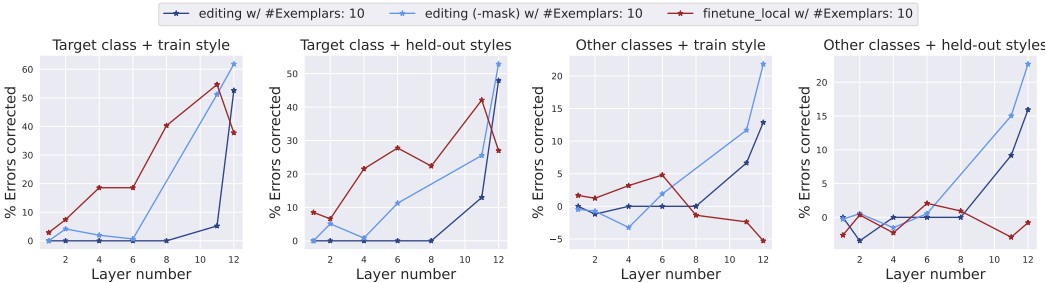

Figure 21: Editing vs. fine-tuning performance (with 10 exemplars) on an ImageNet-trained VGG16 classifier, as a function of the layer that is modified. Here, we visualize the average number of misclassifications corrected over different concept-transformation pairs, with concepts derived from instance segmentation modules trained on MS-COCO; and transformations "snow" and "graffiti". For both editing and fine-tuning, the overall drop in model accuracy is less than $0.25\%$.

**Number of exemplars.** Increasing the number of exemplars used for each method typically leads to qualitatively the same impact, just more significant, cf. Appendix Figures 15-18. We also perform a more fine-grained ablation for a single model (ImageNet-trained VGG16 on COCO-concepts) in Figure 22. In general, for editing, using more exemplars tends to improve the number of mistakes corrected on both the target and non-target classes. For fine-tuning, this improves its effectiveness on the target class alone, albeit the trends are more noisy.

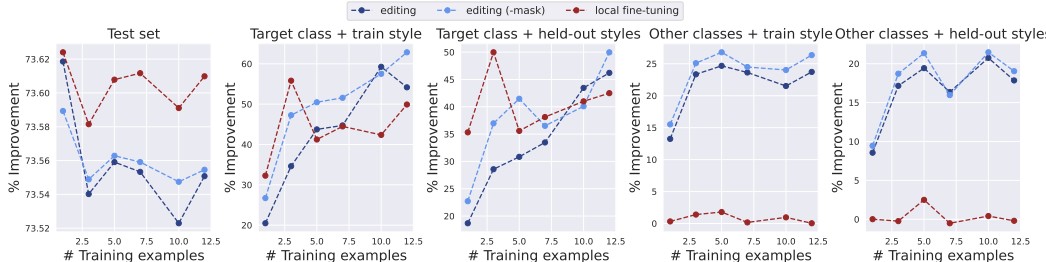

Figure 22: Editing vs. fine-tuning performance on an ImageNet-trained VGG16 classifier, as a function of the number of train exemplars. Here, we visualize the average number of misclassifications corrected over different concept-transformation pairs, with concepts derived from instance segmentation modules trained on MS-COCO; and transformations described in Appendix A.4.1. For both editing and fine-tuning, the overall drop in model accuracy is less than $0.25\%$.

**Rank restriction.** We evaluate the performance of editing when the weight update is not restricted to a rank-one modification. We find that this change significantly reduces the efficacy of editing on examples from both the target and non-target classes—cf. curves corresponding to '-proj' in Appendix Figures 23-26. This suggests that the rank restriction is necessary to prevent the model from overfitting to the few exemplars used.

**Mask.** During editing, Bau et al. [2020b] focus on rewriting only the key-value pairs that correspond to the concept of interest. We find, however, that imposing the editing constraints on the entirety of the image leads to even better performance—cf. curves corresponding to '-mask' in Appendix Figures 23-26. We hypothesize that this has a regularizing effect as it constrains the weights to preserve the original mapping between keys and values in regions that do not contain the concept.

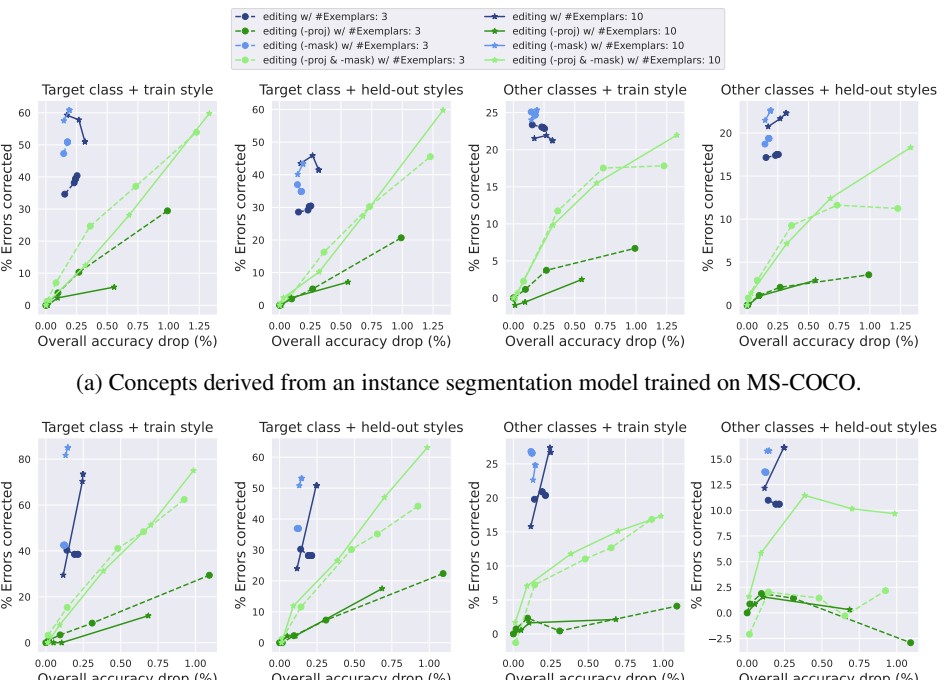

(a) Concepts derived from an instance segmentation model trained on MS-COCO.

(b) Concepts derived from an instance segmentation model trained on LVIS.

Figure 23: Performance vs. drop in overall test set accuracy: Here, we visualize average number of misclassifications corrected by editing variants—based on whether or not we use a mask and perform a rank-one update—when applied to an ImageNet-trained VGG16 classifier.

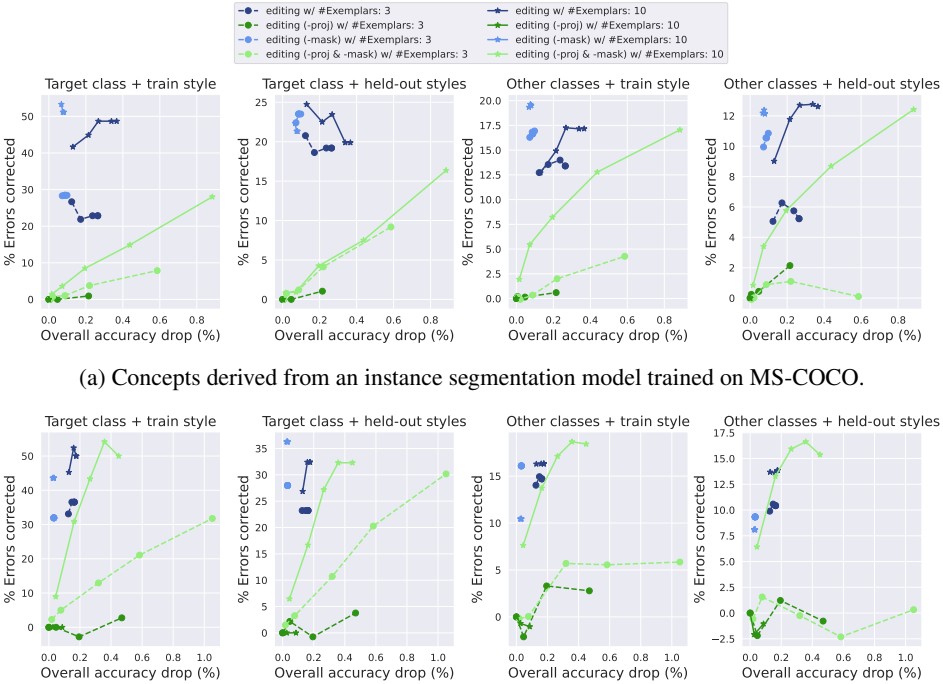

(a) Concepts derived from an instance segmentation model trained on MS-COCO.

(b) Concepts derived from an instance segmentation model trained on LVIS.

Figure 24: Repeating the analysis in Appendix Fig. 23 on an ImageNet-trained ResNet-50 classifier.

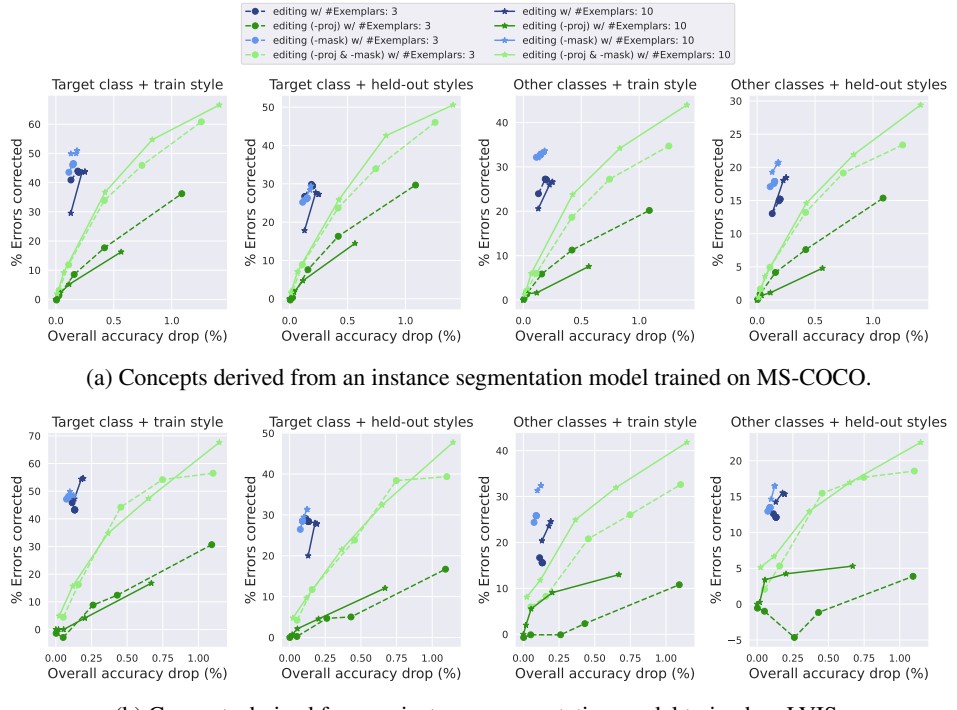

(a) Concepts derived from an instance segmentation model trained on MS-COCO.

(b) Concepts derived from an instance segmentation model trained on LVIS.

Figure 25: Repeating the analysis in Appendix Fig. 23 on an Places365-trained VGG16 classifier.

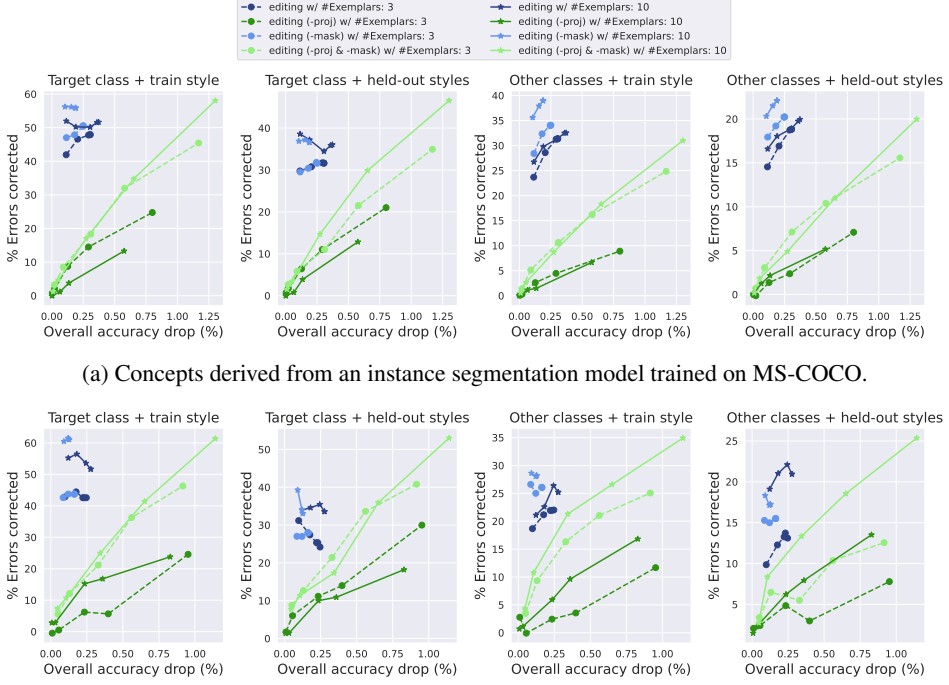

(a) Concepts derived from an instance segmentation model trained on MS-COCO.

(b) Concepts derived from an instance segmentation model trained on LVIS.

Figure 26: Repeating the analysis in Appendix Fig. 23 on an Places365-trained ResNet-18 classifier.

### B.2 Discovering Prediction-rules

Here, we expand on our analysis in Section 5 so as to characterize the effect of concept-level transformations on classifiers.

**Per concept.** In Figure 27, we visualize the accuracy drop induced by transformations of a specific concept for classifiers trained on ImageNet and Places-365 (similar to Figure 6a). Here, the accuracy drop post-transformation is measured only on images that contain the concept of interest. We then present the average drop across transformations, along with 95% confidence intervals. We find that there is a large variance between: (i) a model's reliance on different concepts, and (ii) different model's reliance on a single concept. For instance, the accuracy of a ResNet-50 ImageNet classifier drops by more than 30% on the class "three-toed sloth" when "tree"s in the image are modified, while the accuracy of a VGG16 model drops by less than 5% under the same setup.

**Per transformation.** In Figure 28, we illustrate how the model's sensitivity to specific concepts varies depending on the applied transformation. Across concepts, we find that models are more sensitive to transformations to textures such as "grafitti" and "fall colors" than they are to "wooden" or "metallic".

**Per class (prediction-rules).** In Figure 29, we provide additional examples of class-level prediction rules identified using our methodology. Specifically, for each class, the highlighted concepts are those that hurt model accuracy when transformed.

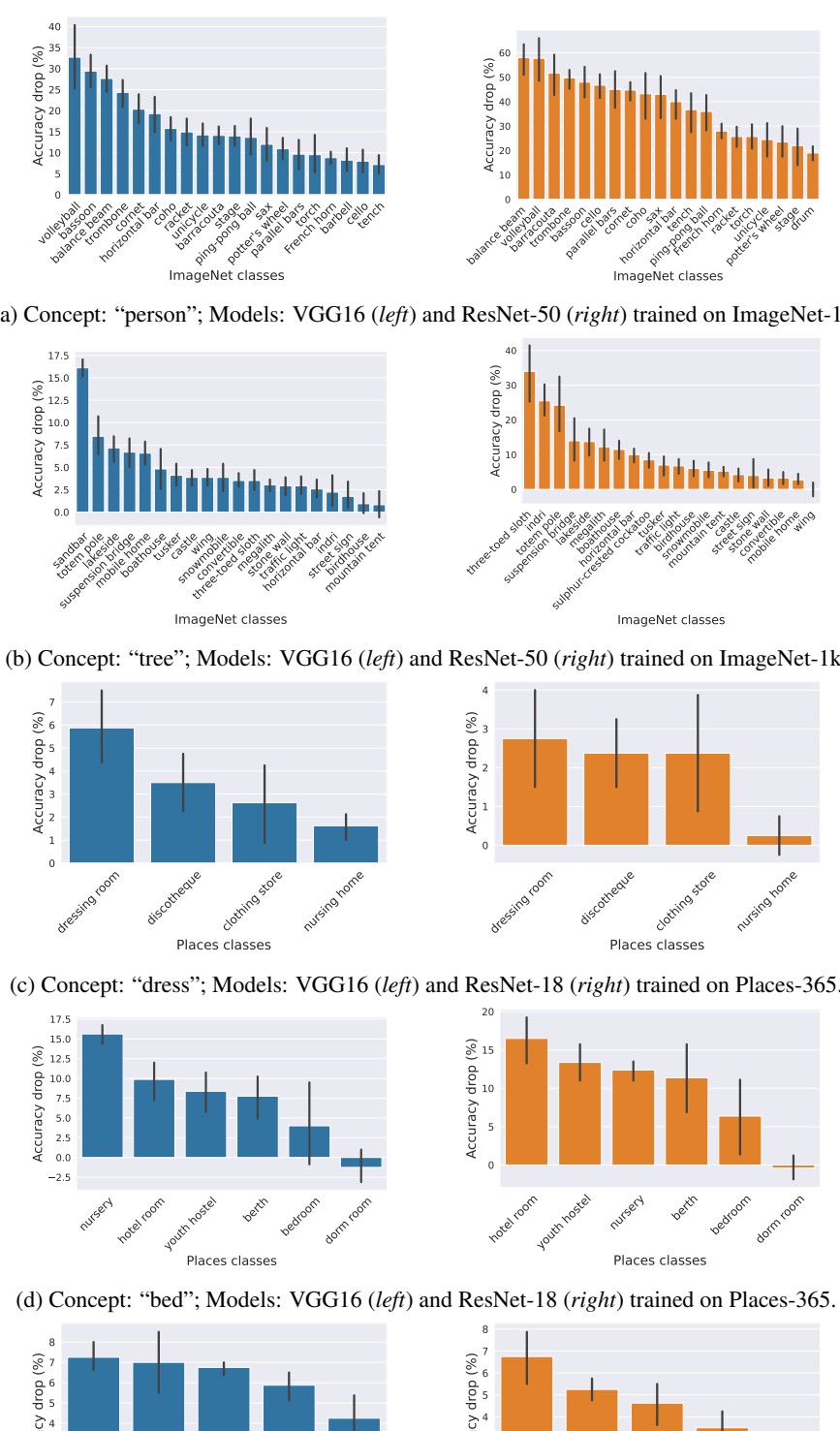

(a) Concept: "person"; Models: VGG16 (*left*) and ResNet-50 (*right*) trained on ImageNet-1k.

(b) Concept: "tree"; Models: VGG16 (*left*) and ResNet-50 (*right*) trained on ImageNet-1k.

(c) Concept: "dress"; Models: VGG16 (*left*) and ResNet-18 (*right*) trained on Places-365.

(d) Concept: "bed"; Models: VGG16 (*left*) and ResNet-18 (*right*) trained on Places-365.

(e) Concept: "signboard"; Models: VGG16 (*left*) and ResNet-18 (*right*) trained on Places-365.

Figure 27: Dependence of a classifier on a specific high-level concept: average accuracy drop (along with 95% confidence intervals obtained via bootstrapping), over various styles, induced by the transformation of said concept. The classes for which the concept is most often present are shown.

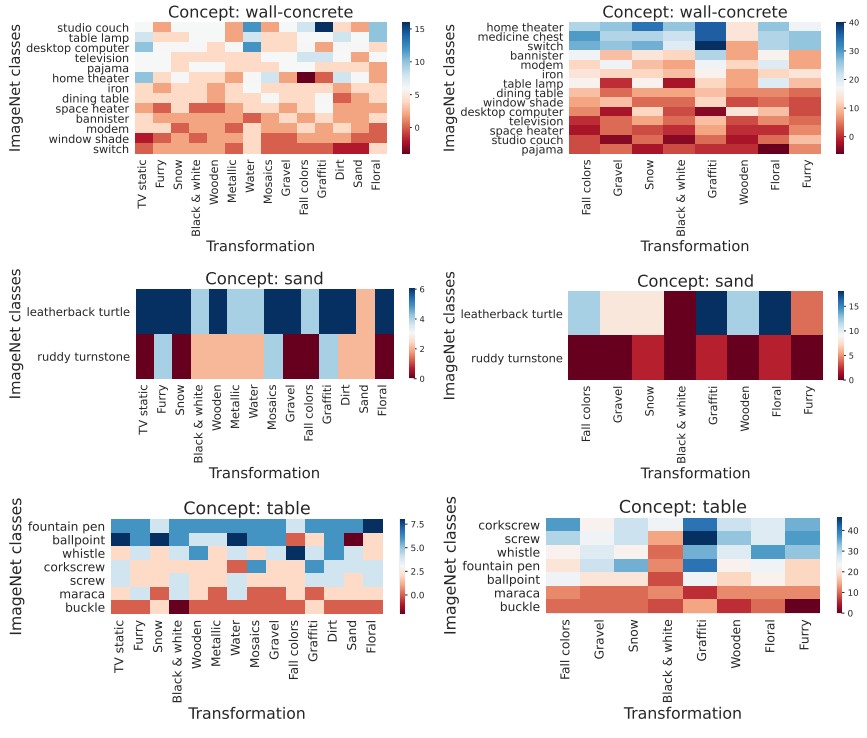

(a) VGG16 (*left*) and ResNet-50 (*right*) models trained on the ImageNet-1k dataset.

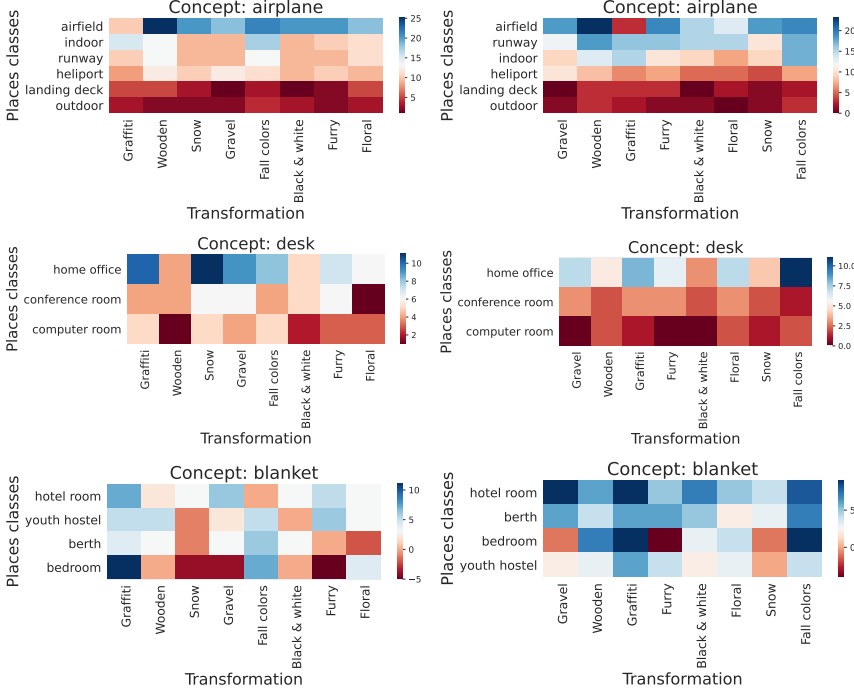

(b) VGG16 (*left*) and ResNet-18 (*right*) models trained on the Places-365 dataset.

Figure 28: Heatmaps illustrating classifier sensitivity to various concept-level transformations. Here, we measure model sensitivity in terms of the per class drop in model accuracy induced by the transformation on images of that class which contain the concept of interest.

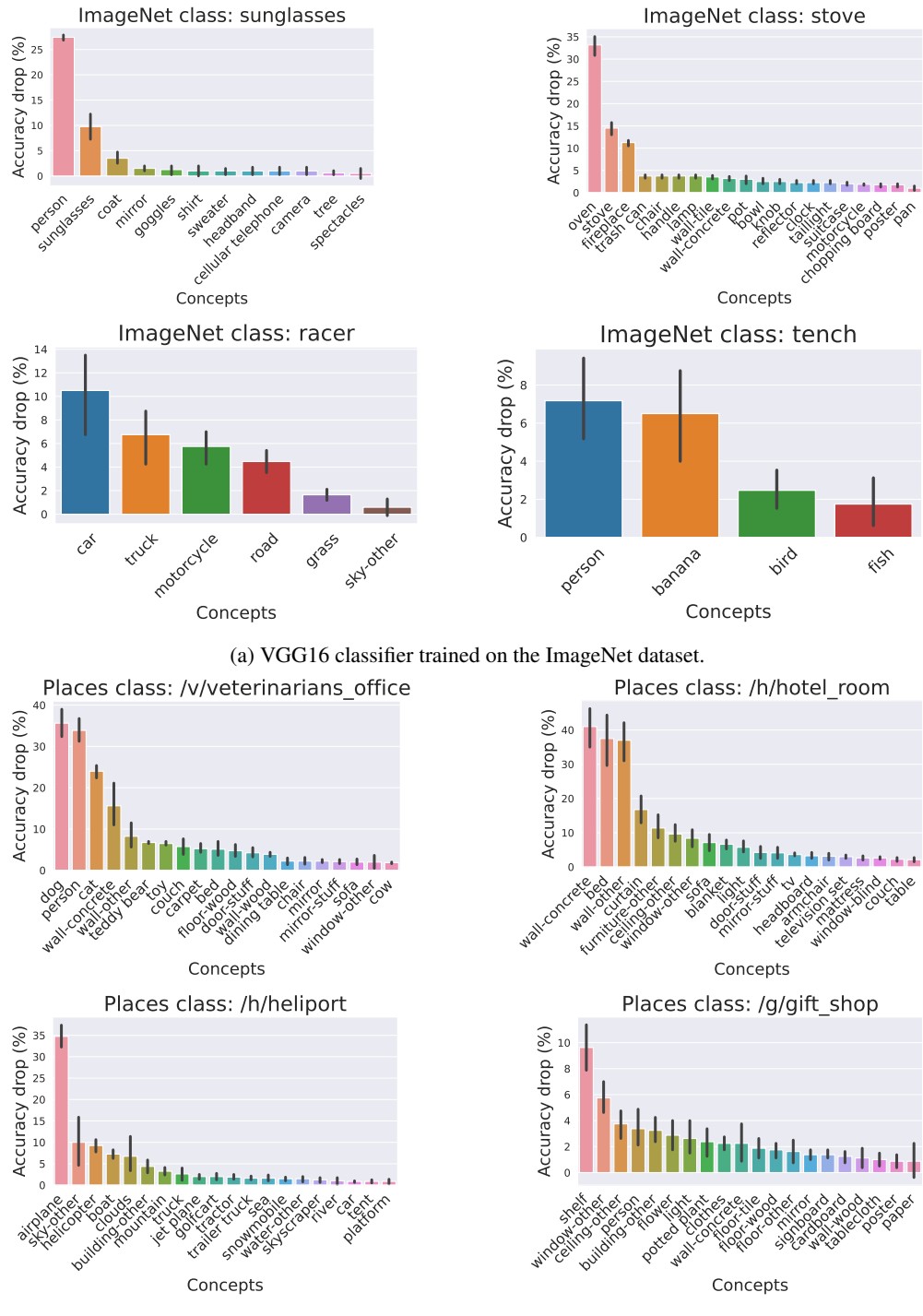

(a) VGG16 classifier trained on the ImageNet dataset.

(b) ResNet-18 classifier trained on the Places-365 dataset.

Figure 29: Per-class prediction rules: high-level concepts, which when transformed, significantly hurt model performance on that class. Here, we visualize average accuracy drop (along with 95% confidence intervals obtained via bootstrapping) for a specific concept, over various styles.