# OpenReview forum: "Editing a classifier by rewriting its prediction rules"
_NeurIPS.cc/2021/Conference — NeurIPS 2021 Poster_

### Official Review · Reviewer_V8H7 · 2021-07-16

**Rating:** 7
**Confidence:** 4

**Summary:**

This paper proposes a method to directly edit the predictions of a trained neural network by (1) building “counterfactual” images by replacing objects in a scene, then (2) mapping activations from pre-transformed images to post-transformed images through the algorithm proposed in Bau et. al. The method is applied to well-known image datasets to show improvements in accuracy on images where transformed objects are present. In particular, the proposed approach generalizes better than naive baselines of fine-tuning on new data. The authors then apply the method to two real world contexts: driving scenes and typographic errors.

**Limitations And Societal Impact:**

I found the limitations and broad impact section to be vague and should be expanded upon. The amount of “manual intervention” and “domain expertise” should be quantified. I mentioned this above but I would like to see effects from the (1) quality of the segmentation model, (2) quality of the counterfactual images. The “broader implications” of direct model editing should also be elaborated upon. For example, what does this enable a malicious actor to do? What kinds of vulnerabilities or new adversarial examples are now possible?

**Main Review:**

Originality: Although none of the individual pieces of the method are new (there are works that replace objects in an image esp. for data augmentation, and much of the method relies on Bau et. al.), I find the method as a whole novel as it enables a new kind of interpretability: users can now audit and correct neural network predictions in a way they could not before.

Quality: I found the ImageNet and Places results to be compelling but the real world experiments less so. I think self-driving cars to be a compelling application of the method but the experiments on vehicles-in-snow represents one small sub-problem. It would be interesting to show more use-cases for self-driving cars. Similarly, I think mitigating spurious correlations is an important problem and one of the motivations for this method (in the introduction). But I found the experiments to be rather simple, and focus on typographic errors. It would be nice to see a larger-scale and more systematic evaluation e.g. Waterbirds dataset.

Clarity: The writing quality was good and I found it easy to understand most of the ideas presented. There were areas I got confused as a reader (see questions below for details). Also, the presentation of Bau et. al.’s method could be more clear, especially to readers (like myself) who are not familiar with the original method.

Significance: Yes, I find the lack of interpretability in neural networks one of the big bottlenecks to widespread adoption. Although the method does not provide an explanation of what neural networks are doing, it does provide a compelling method to correct wrong predictions: a step in the right direction.

Questions on Section 2:
- In generating counterfactuals, how do you choose on what axis to modify an input? For instance, in Figure 2, the tree is modified from a snowy forest to a fall forest. But you could have modified it to replace trees with buildings, or something else altogether. How does the user choose the correct (or best) counterfactual?
- I also wonder how generalizable prediction rules are. A rule like “cars have wheels” is likely a loose approximation of the highly nonlinear decision boundary that the neural network has learned. For some images of cars, perhaps the network relies on wheel features whereas on others, it does not (could be texture / style features). Is there a way to measure how “faithful” my prediction rule is to the neural network?
- I imagine the success of probing model behavior relies heavily on the quality of the instance segmentation. Is it somewhat circular to rely on pretrained segmentation models to identify concepts, which themselves have spurious correlations (that need to be edited out). Is this a chicken and egg problem?
- Similarly, how important is it for the data distribution used to train a segmentation model to be “aligned” with the training data distribution? I imagine if there is a large mismatch, poor segmentation affects the rest of the pipeline. Would be nice to see how sensitive this algorithm is to distribution differences.
- In Figure 3 experiments, how do you pick what a class is transformed to? Specifically, in the example that transformations of “grass” affect accuracy of predicting “croquet ball”, what does “grass” get replaced by? A random object from a random image?
- In Figure 3, what does style mean? Why does changing style affect accuracy much more (~20% vs ~5%) than changing concept?
- How much of the drop in classification accuracy in Figure 3 is due to counterfactuals being out of domain (e.g. looking very different from training distribution), versus detecting model sensitivities? For instance, replacing “grass” with “megalith” likely creates an image that is fairly out of domain, and which the model should not do well on.  How do you differentiate between these two kinds of errors? Similarly, when replacing  objects, do you have to be conscious of lightning, camera angle, resolution, object size, etc. Otherwise, you may run into OOD problems again. How do you filter out cases in which the counterfactual is not (visually) reasonable?
- It makes sense that croquet ball and grass are correlated and hence performance suffers. But I might also expect this for “wood rabbit” (e.g. rabbits usually appear on grass) where it is affected much less. Why is this? What is the relationship between feature correlation and accuracy drop?

Questions on Section 3:
- Is S in Equation 1 derived from the segmentation mask?
- I like the use of Bau et. al. [5] for editing classifiers by key-value replacement. Clever!
- Personally, I am not extremely familiar with Bau et. al. and would love some exposition on why it is that the method can replace objectives without changing the model’s behaviors in other contexts. How is this latter statement guaranteed?
- How do you optimize Equations 1 & 2? In particular, how do you compute k*_ij and how do you enforce the constraint (e.g. Lagrangian?). Some discussion on the optimization complexities would be useful.
- In figure 4’s caption, it is stated that the weights W are updated to enforce a new key-value association. This makes sense but when adding multiple key-value associations, do you have to optimize W to enforce them all at once? I imagine you cannot do this sequentially as W may no longer enforce previous associations when optimized to satisfy a new association. If this is true, does it get very difficult to enforce more and more associations? Are there sets of associations in which a single weight W cannot satisfy?

Questions on Section 4+5:
- Results in Figure 5 are very cool. It makes sense that the finetuning outperforms editing on the target class. It was nice to see that editing generalizes better to unseen examples with the transformed concept whereas fine-tuning overfits. The observation that methods generalize more easily to different styles is interesting.
- I wasn’t clear on the single synthetic exemplar description: does this mean editing was applied in a one-shot scenario? Does this mean fine-tuning is done on a single example as well? If so, it would be interesting to finetune on more examples. For example, is it possible to show that editing is as good as fine-tuning on N new data points?
- I think the iPod example is really cool but would like to see a larger study on a more well-known dataset of spurious correlations. (In particular, Figure 6b suggests that there are only ~6 images?). Would be compelling and interesting to apply this method to the Waterbirds dataset, which has more subtle spurious correlations.

Nit:
- Typo on page 9 line 270 (“neurons” duplicated).


**Time Spent Reviewing:**

5

---

> ### Author Response · Authors · 2021-08-10
> **Author response**
>
> We thank the reviewer for their comments and suggestions.
>
> [Variety of real-world demonstrations] The size/variety of our test sets in Section 5 was constrained by the availability of relevant de-identified data on publicly available sources (e.g., with the creative commons license on Flickr). We thus supplement our practical demonstrations with a large-scale (albeit synthetic) benchmark created using our rule-discovery pipeline (from Section 2) on which we can benchmark our approach at scale and for a range of test conditions (different concept-style pairs) in Section 4.
>
> [Waterbirds dataset] We thank the reviewer for their suggestion of the Waterbirds dataset and would be happy to incorporate it in the manuscript revision. That said, Waterbirds is similar to the benchmarks created using our rule-discovery pipeline in Section 4. In particular, both are synthetic datasets---our benchmarks are created using style transfer, while Waterbirds is based on background replacement. Additionally, both datasets focus on replacing one concept in the image with another---”land” with “water” in Waterbirds, and concepts such as “grass” or “road” with “gravel”/”snow”/”flowers” in ours (see Appendix Figure 21 and 22 for some examples).
>
> ### Questions on Section 2:
>
> [Axis for counterfactuals] In general, the choice of axis is likely to be determined by the application of interest. For instance, to enable the vision system of a self-driving car to generalize from San Francisco to Alaska, the pertinent counterfactual might be “snowy road -> road”; to eliminate typographic attacks it would be “iPod text -> blank”. In some cases the choice of the pertinent counterfactual might be apparent to the user based on model failures in practice (e.g., in the case of typographic attacks), while in others it might require probing the model via interpretability studies (e.g., using the rule-discovery pipeline in Section 2). We thus analyze the performance of our method for a variety of realistic counterfactuals in Sections 4 and 5.
>
> [Faithfulness of prediction rules] We view our analysis in Section 2 as one possible technique for answering this question. In particular, here, we measure how the model behaves when a single concept (say “grass”) in dataset images is transformed (say replaced with “gravel”) using style transfer. By doing so, we can not only measure how reliant the model as a whole is to the feature of interest (here “grass”), but also identify the specific images/classes that are particularly sensitive to this feature.
>
> [Quality of instance segmentation] We agree with the reviewer that the underlying instance segmentation models may have their own biases and artifacts. However, we find that our model probing pipeline, while not perfect, is already effective at identifying and quantifying unintended model behaviours at scale. Moreover, further improvements in pre-trained segmentation models will only strengthen our approach.
>
> [Distributional differences] Note that the segmentation models used for our analysis are trained on different distributions (COCO and LVIS) than the classifiers (ImageNet and Places). In fact, we view this as a core strength of our approach---namely, we can leverage a pre-existing segmentation network based on a different dataset, as long as it can detect concepts relevant to the classifier. We view a deeper analysis of the effect of distributional differences on performance as a valuable direction for future work.
>
> [Figure 3; transformations] The concept transformations in Figure 3 are based on the two steps outlined in Section 2.1. In particular, each concept (e.g., “grass”) is transformed using style transfer with respect to a selected style (e.g., “gravel” or “snow”). We focus on a set of 14 realistic styles detailed in Appendix A.4. In Figure 3, we report the average effect of replacing grass with each of these 14 styles on model accuracy.
>
> [Figure 3; style] As discussed above, the style refers to the transformation applied to a concept via style transfer. For instance, in Appendix Figure 7, we illustrate examples of ImageNet images with the concept “road” stylized using each of the 14 styles. Note that Figure 3a measures the drop in model accuracy for specific ImageNet classes when “grass” in the corresponding images is transformed, averaged over different styles. On the other hand, Figure 3b measures the drop in overall model accuracy, averaged over classes and concepts, for each style. Thus the drop across the two images is not directly comparable.
>
> [Figure 3; out-of-domain] To clarify, we never replace a concept with a random object, but instead transform it using style transfer with respect to a set of pre-selected styles. These styles have been chosen to capture transformations that are likely to arise naturally (see Appendix Figure 7 for some examples). Further, as we discuss in A.6.1, we manually ensure that the transformed concept (say grass) does not significantly obscure the object of interest (say “croquet ball”), as this would otherwise preclude correct classification. Overall, because we take care to (i) not modify the class object and (ii) use style transfer that is typically fairly realistic, we expect the resulting images to not be grossly out-of-distribution.
> Further, we see evidence that the drop in model accuracy is due to its reliance on the concept and not due to a gross distribution shift because it tends to be highly non-uniform across classes (Figure 3a and Appendix Figure 9). For instance, even in classes where “grass”  is highly prevalent  (Figure 3a), its transformation (averaged across 14 styles) affects “croquet ball” ~30% more than “collie”.
>
> [Feature correlation and accuracy drop] The classes shown in Figure 3a (and Appendix Figure 9) all have similar feature correlation---comparable number of test set images contain “grass” in a comparable fraction of pixels. (In particular, as described in the feature caption, we specifically choose to illustrate 20 classes where the concept “grass” is most frequently present.) Despite this, we see that the model accuracy drop across these classes is highly non-uniform. We will make a point to clarify this in the manuscript revision. A deeper analysis into why model reliance on features does not directly mirror the feature correlation in the dataset would be a valuable direction for future work.
>
> ### Questions on Section 3:
>
> [S in Equation 1] Here, S denotes a binary mask denoting the concept of interest. This mask can either be obtained using pre-trained segmentation models (as in Section 2) or annotated manually (as in Section 4).
>
> [Bau et al. for editing classifiers] Thank you!
>
> [Exposition of Bau et al.] Bau et al. enforce this by solving a constrained least squares problem---to introduce a specific key-value association between $k^*$ and $v^*$, while regularizing the weight to preserve the mapping between other key-value pairs in the dataset. This amounts to constraining the change in the weight (via projected gradient descent) to be along a specific rank-1 direction. We will include a more detailed discussion of this in the revised manuscript.
>
> [Optimizing equations 1 and 2] To compute $k^*_{ij}$, we obtain on the representation of the image at the *input* of the layer being edited via forward propagation. We then use projected gradient descent to minimize the least squares loss (1), to ensure that the change in weights lies along the rank-1 direction $C^{-1}d$. Due to space constraints, and since this part of the method is identical to the one of Bau et al. and discussed at length in their paper we opted to omit it. Still, we would be happy to include a short summary.
>
> [Multiple key-value updates] It is not entirely clear what is the best way for implementing multiple simultaneous rewrites. For instance, one could consider a rank-N update when editing N rules simultaneously. Alternatively, one might also hope that as long as these rule-rewrites are not in conflict with each other, then performing them sequentially (with rank-1 updates) might also work. We believe that this is an important direction for future research.
>
> ### Questions on Section 4+5:
>
> [Results in Figure 5] We thank the reviewer for their comment!
>
> [Single synthetic exemplar] Yes, here both editing and fine-tuning are based on a single exemplar. We observed that the performance gap between editing and fine-tuning did not reduce (and for some classes increased) using additional exemplars. In particular, similar to Appendix B.2.3, we found that while with additional exemplars (N=1-10):
> - Both editing and fine-tuning performance improves on the target class.
> - On all other classes, editing performance improves much more drastically than fine-tuning (which often remains unchanged).
>
> We will include these results in the Appendix.
>
> ### Questions on limitations and broader impact:
> [Manual intervention and expertise]
> First, we want to clarify that domain knowledge and domain expertise as a prerequisite is not specific to our method but rather to any method where a human designer interacts with a model. That being said, quantifying the amount of domain expertise necessary for any such method is tricky. For instance, in our setting, the classification of both “groom” and “suit” might be sensitive to the presence of a “jacket”. However, depending on the application, one might want to modify the former dependency and not the latter.
>
> [Role of a malicious actor]
> Just as direct model editing can be used to correct for undesirable model behavior, it can be used to introduce malicious behavior into a model. For instance, it can be used to discriminate against a group of users (by mapping a group-specific attribute to an undesirable prediction) or cause the model to catastrophically fail in certain situations (e.g., “snow -> accelerate” in self-driving cars).

---

### Official Review · Reviewer_4Ajg · 2021-07-16

**Rating:** 6
**Confidence:** 4

**Summary:**

The paper presents a method to improve robustness of object classifiers to style-changes in components, termed “concepts” in the paper, which can include both background elements as well as parts of the object of interest. This is done by first localizing confounding concepts using existing instance segmentation methods, performing style-transfer on the localized concept using existing methods, and then using an existing method which performs a rank-1 update to (a layer of) network parameters to map the transformed-concept-region to the same value as the original concept-region was mapped to by the layer. Experiments illustrate that this pipeline is effective at making the network robust to style-changes for concepts identified to be potentially misleading.

**Limitations And Societal Impact:**

Limitations and potential societal impact have been adequately discussed.

**Main Review:**

The objective of the proposed toolkit seems potentially useful: as the paper says, “after identifying undesirable prediction rules, a user might want to modify them before deploying their model in the wild”. Synthetic experiments indicate that it is possible to significantly improve robustness to style-changes, both for the trained transformations as well as held-out styles.

It seems that the setup being considered is one where one does not have the budget to re-train (otherwise a baseline that emerges naturally is to augment the training set, perhaps on the fly, with the style-transferred examples). Perhaps this point should be motivated and made clearer somewhere in the paper.

Two realistic examples of the toolkit-in-action are demonstrated: using a synthetic exemplar of a vehicle on snowy roads, improved generalization to vehicles on actual snowy roads is obtained; a possible application to avoid typographic attacks is shown, by editing out the specific typographic attack using counterfactual images. While these are interesting demonstrations, the test sets seem to be on the smaller side - the snowy roads test set seems to contain around 120 examples, and the typography-attack set seems to consist of 6 (x2) pictures. One way to make the toolkit developed in the paper more compelling might be to showcase improvements in challenging test-sets such as Imagenet-A [1], where it is suggested that at least some of the images exacerbate generalization failure due to over-reliance on specific features.

For the fine-tuning baseline, was there a reason the number of steps was capped at 800? Could performance improve with more steps? (Especially given that the number of optimization steps for the proposed method seems capped at a much higher 80000.)

Minor suggestions:
Perhaps “concept localization” might be a better term instead of “concept identification” in L69; at first glance I interpreted it to mean the process of discovering misleading concepts.

On the whole, I think the paper presents a technique that can potentially be inspirational, or of use to practitioners in some areas. However, some of the underlying assumptions (i.e. no re-training budget, but continued access to data and ability to perform style-transfers) might be too narrow to be realistic. I encourage the authors to add a comment or two describing the specifics of situations where such a pipeline would be a natural choice, to help the paper find its right audience.


[1] Natural Adversarial Examples, Hendrycks et al., 2019

------------
I have read the rebuttal, and stand by my rating.

**Time Spent Reviewing:**

6

---

> ### Author Response · Authors · 2021-08-10
> **Author response**
>
> We thank the reviewer for their comments and suggestions.
>
> [Relevant settings and assumptions] Our focus is indeed on the setting where re-training a model from scratch may not be feasible---e.g., due to limited training data or computational resources. These constraints are fairly likely to arise in practical applications, such as self-driving cars or medicine, where one might want to adapt a model to a new environment (e.g., different country or hospital) or modify specific decision rules learned by it (e.g., eliminate reliance on weather conditions or visual artifacts). Moreover, note that the CLIP model edited in S5 is precisely an example of a publicly released model that has been trained on a proprietary dataset using a large computational budget.
>
> Additionally, note that our editing approach does not assume access/ability to perform style transfers. In particular, the rule-discovery pipeline (based on style transfer) in Section 2 and the editing pipeline in Section 3 can be treated as  standalone components, and the former not a prerequisite for the latter. In Section 4, we leverage the rule-discovery pipeline to create a benchmark to quantify the efficacy of editing/fine-tuning decision rules at scale. However, in practice, akin to the demonstrations in Section 5, one could simply rely on a few manually-created exemplars for training, and then evaluate the edited model on naturally arising test cases (e.g., real vehicles on snowy roads).
>
> We will clarify this in the manuscript revision.
>
> [Test set size] The size of our test sets in Section 5 was constrained by the availability of relevant de-identified data on publicly available sources (e.g., with the creative commons license on Flickr). We thus use the rule-discovery pipeline in Section 2 to create a large-scale (albeit synthetic) benchmark in Section 4 on which we can benchmark our approach at scale and for a variety of test conditions (different concept-style pairs). On the suggestion to utilize ImageNet-A:
> - Although spurious patterns (e.g., wood ->“nail”, feeders -> “hummingbird”) have been observed in ImageNet-A images, to the best of our knowledge, there is no deterministic evidence that these specific patterns are responsible for model errors. Thus, a priori, it is not obvious that ImageNet-A failures can be corrected by rewriting the decision rule mapping a particular spurious pattern (say wood) and a class (say “nail”).
> - Moreover, while the ImageNet-A test set itself is large, the number of  examples that the model gets incorrect due to a given decision rule (e.g., wood ->“nail”) is likely to be comparable to the size of the test sets we consider in our work.
>
> Nevertheless, we agree that applying editing to rectify ImageNet-A failures could be a valuable direction for future work.
>
> [Fine-tuning steps] The hyperparameter ranges for each of the methods were determined based on an exploratory grid search. In general, we observe that the performance of fine-tuning saturates after a few hundred steps. In contrast, editing requires more iterations---presumably due to the rank-1 projection at every iteration.
>
> [Concept localization] We thank the reviewer for pointing this out---we will rephrase L69 as concept localization instead.

---

### Official Review · Reviewer_MsmY · 2021-07-19

**Rating:** 4
**Confidence:** 4

**Summary:**

The paper proposes an algorithm to change the behaviour of a trained model in the presence of some external interference in the presence of an external concept. The paper does it without using extra data in two stages. First, the algorithm utilises a rule detection framework using counterfactuals and then changes it using the technique from bau et. al.

**Limitations And Societal Impact:**

yes

**Main Review:**

## Originality
* The paper is largely based on existing works including Bau et. al. [5]. Bau et. al [5] work with generative models whereas the authors here look at classification models. But, the main technique of how to insert a new prediction rule is almost exactly similar.

## Significance

* There is very little systematic quantification of the success of the algorithm. For example, when style editing is done on a large scale on all images of a class for one concept, does it make sense to expect a correct classification of the transformed images ? Does the style/concept editing actually produce proper images for all ? I am not sure what a systematic way to measure success would be but this lack of a proper way to do a systematic study probably originates from the absence of a proper formulation of the problem. While there doesn't need to be a theoretical understanding, I think there needs to be a rigorous problem formulation regarding what are _concepts_ what are _rules_ and what this method is supposed to achieve.
* There are also no explanations for some of the observations eg. In Line 664 in the supplementary the authors state that " the accuracy of a ResNet-50 ImageNet classifier 664 drops by more than 30% on the class “three-toed sloth” when “tree”s in the image are modified, while the accuracy of a VGG16 model drops by less than 5% under the same setup." What makes resnets so vulnerable  ? This seems like a weird phenomenon and should be explored further.
* There is no discussion of the limitations of the algorithm. For example, how many rules can be changed simultaneously ? What kind of rules are easier to change and why ? The _application to real world_ is also a very particular example of using a _snowy_ weather.
* I think some of the claims of the paper should be downplayed a bit. For example in the abstract, the authors state _"including adapting a model to new environments, and modifying it to ignore spurious features"_. But the paper doesn't show how to detect and remove spurious features (in addition to not actually defining what a spurious feature is". Also, there is no example of adapting the algorithm to new environments (again no definition of what an environment actually is). The abstract makes it sound like the paper actually achieves all of this, which it doesn't, at this state.



Overall, there are a lot of different observations of various scales in the paper which can pose interesting questions. But the efficacy of the algorithm has mainly been demonstrated in the main text through anecdotal examples  and the problem lacks a proper formulation, which I think is necessary to distinguish between success and failure of the algorithm. Thus, I am unwilling to vote for acceptance of this paper at  the level of NeurIPS.

**Time Spent Reviewing:**

6

---

> ### Author Response · Authors · 2021-08-10
> **Author response**
>
> We thank the reviewer for their comments.
>
> [Novelty] While the underlying primitive of key-value replacement is identical to that in the work of Bau et al., their focus is on introducing objects in generated images, while ours is on rewriting a classifier’s decision rules. These objectives are fundamentally different, and achieving the latter requires us to put forth a novel conceptual framework---e.g., identifying the forms of classification errors that can be corrected using it, as well as designing benchmarks and baselines to evaluate it. Also note that in contrast to Bau et al., we are able to quantitatively measure the effectiveness of key-value replacement (in prior work the evaluation is purely done qualitatively by inspecting the generated images), which allows us to perform a detailed investigation into the role of various algorithmic design choices (Appendix B.2.3). More broadly, to the best of our knowledge, we are the first to study the problem of editing decision rules of a pre-trained classifier in a targeted manner.
>
>
> [Rigorous evaluation] We want to emphasize that we do not propose a single monolithic pipeline but rather two complementary, yet standalone components: the rule-discovery pipeline that enables one to identify potential problems; and the editing methodology for addressing those problems.
>
> The rule-discovery pipeline is at its core a model debugging tool. Fully evaluating usability of debugging with human subjects can be challenging and is beyond the scope of our work.
>
> With regards to our editing method, we perform extensive experimental analysis using a well-defined metric (3) to characterize its effectiveness. In addition to our practical demonstrations in Section 5, we corroborate our results at scale under a wide-range of conditions using a test bed created via the rule-discovery pipeline (section 4.2 and supplemental section B.2.1, B.2.2, B2.3.3). In particular, this pipeline allows us to create a suite of test cases wherein the transformation of a single concept in dataset images, based on a specific style, greatly impairs models. We then quantitatively demonstrate, across datasets (ImageNet and Places) and architectures (VGG and ResNet50), how editing improves model generalization in these settings.
>
> [Should we expect correct classification when transforming concepts?] Our tool systematically demonstrates many instances in which classification accuracy drops when a concept in the image is transformed (supplemental fig 9, 10, 11). However, not all such drops are undesirable: the importance of any such sensitivity in the model depends on how it will be deployed. The goal of a model debugging tool is to expose the behavior of the model to a human designer who can then decide which parts of that behavior are undesirable for their application. For instance, the reviewer mentions that ResNet50 and VGG16 respond very differently to transformations of trees when recognizing sloths. This demonstrates that VGG16 models rely less on trees which might be desirable in certain situations or irrelevant in others.
>
> [Formalizing concepts and rules] While the notion of a “concept” and “prediction rule” is fairly intuitive and commonly-discussed in machine learning, they can be quite challenging to formalize. What we view as a key strength of our framework is that it allows us to specify and manipulate these quantities implicitly through data. For instance: .
> - In the context of the rule-discovery pipeline concepts are defined through the dataset used to train the segmentation model. That is, although it is hard to mathematically define “grass”, one can leverage humans to annotate that concept in images.
> - In turn, prediction rules are relatively intuitive sentences of the form "The accuracy of the model drops by X% on class Y when we replace 'road' with 'grass'." In our editing pipeline, we allow users to edit such prediction rules by implicitly specifying the model behaviour (treat “wooden wheels” like standard ones) via exemplars.
>
>
> [Analysis of the algorithm]
> - In Appendix B.2.3, we perform a detailed analysis of how various algorithmic design choices (using masks, number of examples, choice of layer) affect performance.
> - In Appendix Figures 21 and 22, we perform a comparative study of which concepts and transformations are easier to edit.
> - We did not experiment with rewriting multiple rules simultaneously since we believe that there is significant depth there, which would be better explored in future work.
>
> [Downplaying claims] We do not agree with the reviewer that our abstract is overclaiming our contributions. In terms of adapting to new environments, we present a large set of experiments on synthetic datasets (Sec 4) and real images (snow in Sec 5) where we improve the performance of a model when different aspects of the data distribution change---e.g., roads are covered with snow, grass is replaced with wood, etc. In terms of spurious correlations, we study typographic attacks in Section 5.  In particular, it has been observed that CLIP models spuriously associate the text “iPod” with iPods---after all, while most iPods tend to have the text on them but simply writing "ipod" on a random object does not make it an iPod. We show how editing can significantly reduce model reliance on such spurious text.

---

### Official Review · Reviewer_VmAm · 2021-07-21

**Rating:** 7
**Confidence:** 4

**Summary:**

The authors propose a model editing method that proceeds in two steps. First, they rely on off-the-shelf segmentation models and style transfer models to learn a data transformation $x \to x'$ that preserves semantic meaning (e.g., in a vehicle recognition task, they might convert a snowy road into a non-snowy road, which preserves the label of the vehicle). Second, they build on recent work by Bau et al. to use these transformed pairs $(x,  x')$ to learn an edit to the model $f$ such that, roughly speaking, $f(x)$ and $f(x')$ agree in their representations at some particular layer and above. They show that their procedure generalizes (astonishingly) well from edits learned on just a single exemplar.

**Limitations And Societal Impact:**

Yes.

**Main Review:**

I found the paper interesting and well-written. Thank you to the authors for this submission! While each component of their pipeline is not "new", in the sense that they adopt the model editing method from recent work and use off-the-shelf models otherwise, the overall idea and their analysis is still interesting and novel. I think this will be a positive contribution to the research community.

1. My main feedback is around the framing of the problem setting and how the method requires "essentially no additional data collection" (L30) or "any annotation effort" (L81). Related to this concern is that there are potentially stronger baselines that could be run against the authors' model editing procedure.

Concretely, it seems that the authors consider two related but distinct settings:

a) In Sections 2 and 4, the authors describe an automated setting that relies on off-the-shelf segmentation models to segment different concepts ("grass", "sea") and then off-the-shelf style transfer models to transform images according to these concepts (e.g., converting grass in one image to sea). Once the library of concepts and the set of acceptable transformations are established, this is an essentially automatic process that can be scaled to multiple images.

b) In Sections 5, the authors describe a manual setting where they use prior knowledge of the test distribution to construct a single example (pair) that can then be used to learn a model edit.

These settings are a bit conflated in the paper, but they seem distinct. In particular, if we are in the scalable setting (a), where we're learning transformations through style transfer, then it is a bit unclear to me why we are restricting ourselves to experiments in Section 4 with a small number of exemplars ($N$ is not specified in the main text, I think, but from the appendix it seems to be $N=3$ for Figure 5) and with exemplars from only one class. The authors show that in this setting, model editing outperforms finetuning (for example, the performance gains from model editing extend to other classes), but it's unclear to me why we're in this setting in the first place. If we can just generate more examples from more classes using the same automated rules, then from Figure 5, it seems like finetuning could perform significantly better than model editing.

Besides finetuning, another natural baseline would also be to use the learned data transformation (e.g., grass -> sea) and then plug it in to any number of contemporary methods that use these kinds of data transformations, such as contrastive learning, consistency regularization, or even plain data augmentation etc. It'd be a bit surprising if those methods didn't outperform model editing given sufficient data. If there's a reason why it's hard to get "sufficient data" in setting (a), for example because it's computationally too intensive to construct these transformed images or something like that, that would be good to discuss.

Setting (b) seems like a more compelling demonstration of model editing, since it only relies on a single exemplar. But this setting is introduced only later in the paper, and doesn't seem to be the focus of the first few sections. Even in Section 5, it was unclear why we were in setting (b), since if we come up with the rewrite rule snow -> road, it seems like it'd be straightforward to generate many more than one example. What happens when we have more than one example?

Overall, could the authors clarify precisely what constraints and problem setting we're operating in, the motivation for it, and the reasoning for the baselines chosen?

---

Other minor comments:

2. The "no manual annotation" claims could be more nuanced. First, the off-the-shelf models were trained on extensive manual annotations, so this method only works "without manual annotations" on datasets where good off-the-shelf models have already been trained (with manual annotations). Second, identifying a reasonable set of concepts (e.g., disallowing the dog concept from being transformed, as the authors note) is also a form of manual annotation, albeit much less labor intensive.

3. It might be worth discussing work in the interpretability literature that learn models which explicitly encode concepts. Examples of this include
recent papers on concept whitening models (https://arxiv.org/abs/2002.01650) and concept bottleneck models (https://arxiv.org/abs/2007.04612), which also discuss intervening on concepts but in a different way from the proposed method. Another example is the work on semantic bottleneck models (https://arxiv.org/abs/1907.10882), which consider semantic segmentations as well, and study the effect of intervening on those segmentations. Broadly speaking, these are all similar in the sense of using an existing library of concepts to understand and intervene on model behavior, though the authors' proposed method here is still distinct and novel.

4. The two components of the pipeline (segmentation + style transfer, and then model editing) seem like they are somewhat independent. For example, one could do model editing to remove spurious correlations using external metadata (e.g., if you know the race of the person in a photo and you want to edit the model to map people of different races into the same representation; or if perhaps you know that you want to be robust to backgrounds as in https://arxiv.org/abs/2006.09994). As a suggestion, discussing this might be helpful for broadening the scope of the proposed method.

---
Update: Thanks to the authors for the response. I would like to keep my score. I think the paper contains interesting ideas, though I agree with the other reviewers that the setting could be clarified and the experiments made more rigorous.

**Time Spent Reviewing:**

3

---

> ### Author Response · Authors · 2021-08-10
> **Author response**
>
> We thank the reviewer for their comments and suggestions.
>
> [Regime of interest] We view the few-samples regime as the key application of the editing methodology. Specifically, in many practical settings, one might not have access to a large dataset (especially one with fine-grained concept-level annotations). In such cases, being able to rewrite the desired model behavior with a handful of examples is particularly relevant. Thus, even though we are able to create a large number of exemplars from multiple classes automatically using our rule-discovery pipeline, we only view this as a mechanism to: (a) identify the decision rules to edit and (b) create a large-scale testbed to evaluate editing on.
>
> Moreover, we want to note that even in settings where we have enough data to specify the relevant concepts well (e.g., large datasets containing annotated "roads"), editing these concepts is a much more direct and targeted process than fine-tuning with a lot of data. That is, when fine-tuning, the only signal that we provide to the training process is that these inputs should be classified according to their correct class. This can often lead to the model focusing on other characteristics of the input or even simply manipulating class biases (e.g., see Figure  B.3).
>
> [Minor comments] We agree with all these points and will address them in the revised manuscript. Specifically:
> - The segmentation models used have indeed been trained on annotated data. However, the annotation cost is paid once when creating the respective training dataset. When using these models as part of our rule-discovery pipeline no additional annotation/collection is required. Furthermore, identifying a set of reasonable concepts to modify (e.g., dog concept) is not relevant to the prediction-rule discovery stage. Instead, this is a necessary cost to determine which prediction rules of the network should even be modified (using any technique). We will edit the corresponding sentences to clarify this.
> - The literature on concept-based models is certainly relevant---we will add references and a short discussion.
> - We do view the two components of the pipeline as largely independent. We will edit the manuscript to emphasize this point.

---

### Decision · Program_Chairs · 2021-09-27

**Decision:**

Accept (Poster)

**Comment:**

The authors propose a model to edit a classifier's decision boundary, in order to enforce invariance of the classifier to regions in the image that are known a priori to not change the semantic label (i.e. a car on a road should still be classified as a car on a snowy road). They proceed in two steps: first they rely on segmentation models to isolate concepts (e.g. road) and style transfer to transform such concepts (e.g. road -> snowy road) to obtain augmentations of the input image. Taking inspiration from previous work, they learn a "translation" between inner features of the base classifier corresponding to the concept in the transformed image, to the corresponding features of the original image, effectively making the transformed image behave like the original one. Experiments show that the procedure generalizes well, even when just learning from very few transformed exemplars.

--

Reviewers recognized the potential impact of the proposed approach in real-world scenarios and the strength of the shown results in a harder few-shot generalization setting (the method is robust on modifications for classes held out during training). A reviewer found this work "thought-provoking" and seen "as a start of a line of research that could be fruitful". All reviewers also recognized the main criticisms of this paper, which turn around: (a) the overall lack of rigor in the problem definition; (b) it's rather unclear what are the limitations of this approach and when the approach may fail; (c) the somewhat anecdotal nature of the experiments, which test for specific concept-rule pairs rather than a more systematic evaluation.

Regarding (a), reviewers found the paper a bit hand-wavy at times, for example, there's no definition of what is meant by "rule". Regarding (b), one reviewer found particularly critical that the paper didn't address the question "how many rules can be changed simultaneously?", given that the paper only deals with changing one rule at a time. Authors are strongly encouraged to think about adding an experiment illustrating this point in the final revision, for example, by verifying performance when training multiple rule translations. During rebuttal, I discussed with the authors a specific experimental scenario that can illustrate a clear failure of this model. I hope the results of our discussion will be included in the paper. Regarding (c), a reviewer acknowledged that the lack of large scale systematic evaluation might also be "due to lack of datasets and the nature of the problem being studied". To make evaluation stronger in terms of baselines, I suggest the authors to incorporate also additional standard contrastive/constitency-based regularization methods, as suggested by a reviewer.

Overall, even if reviewers were not unanimous about acceptance, I believe that the results and use cases showcased in this paper can bring excitement and inspire future work and outweigh the criticisms expressed above. I recommend this paper for acceptance. I suggest the authors do their best to address the criticisms of the reviewers (with a special effort for those who were the most negative) and incorporate the aforementioned suggestions in their final version.